# Emergence of Linear Truth Encodings in Language Models

**Shauli Ravfogel**[1]    **Gilad Yehudai**[1]    **Tal Linzen**[1]    **Joan Bruna**[1]    **Alberto Bietti**[2]
[1]New York University    [2]Flatiron Institute

## Abstract

Recent probing studies reveal that large language models exhibit linear subspaces that separate true from false statements, yet the mechanism behind their emergence is unclear. We introduce a transparent, one-layer toy model that reproduces such truth subspaces end-to-end and exposes one concrete route by which they can arise. We study one simple setting in which truth encoding can emerge: a data distribution where factual statements co-occur with other factual statements (and vice-versa), encouraging the model to learn this distinction in order to lower the LM loss on future tokens. We corroborate this pattern with experiments in pretrained language models. Finally, in the toy setting we observe a two-phase learning dynamic: networks first memorize individual factual associations in a few steps, then—over a longer horizon—learn to linearly separate true from false, which in turn lowers language-modeling loss. Together, these results provide both a mechanistic demonstration and an empirical motivation for how and why linear truth representations can emerge in language models.

## 1    Introduction

Recent observations suggest that large language models (LMs) often encode a low-rank linear subspace that distinguishes *true* from *false* statements across a wide range of domains [Azaria and Mitchell, 2023, Burns et al., 2022, Li et al., 2024b, Marks and Tegmark, 2024, Bürger et al., 2025, Orgad et al., 2025]. Specifically, in many layers of the residual stream representation in transformer-based LMs, a linear separation emerges between representations corresponding to true versus false assertions. Moreover, this separation generalizes across domains: there exists a *single* separating subspace such that statements like "$2 + 2 = 4$" (true) and "The capital city of France is Rome" (false) fall on opposite sides of the same separating plane. These findings have sparked interest among practitioners, because they may aid in mitigating hallucinations [Li et al., 2024b, Orgad et al., 2025].

We investigate the emergence of a unified "*truth subspace*"—a low-dimensional linear manifold that cleanly separates true from false statements. Prior work shows (i) that truth-encoding directions generalize remarkably well across diverse tasks and prompts, and (ii) that causal interventions along those directions can steer LMs toward factual or counter-factual completions [e.g. Meng et al., 2022]. Yet we still lack a satisfying answer to two fundamental questions: *why* do such subspaces arise during training, and *how* are they actually computed at inference time?

We address both questions in a single theoretical and empirical framework. For the *how*, we build on the growing understanding of *key–value associative memories* in transformers. Geva et al. [2021] showed that the first linear layer produces key matches—e.g. aligning the prefix "The capital city of France is" with an internal query—while the second linear layer retrieves the associated value, such as the hidden representation of "Paris". Subsequent studies refined the mathematical description of this mechanism and demonstrated its causal role in factual recall and reasoning [Geva et al., 2022b, Bietti et al., 2023, Cabannes et al., 2024b, Nichani et al., 2025]. We hypothesize that a *linear truth code* takes advantage of the memorized factual associations: it emerges as a result of the model *contrasting*

the internal prediction it built with the observed attribute. This results in a different pattern when the two match or mismatch, and is translated into a linearly separable signal.

For the *why*, we propose the **Truth Co-occurrence Hypothesis** (TCH): in naturally occurring text, true statements are statistically more likely to co-occur with other true statements, and falsehoods with other falsehoods. This assumption is closely related to recent "persona" explanations of factual inconsistency in LMs [Li et al., 2023a, Joshi et al., 2024]: the claim that LMs learn to model certain personas in the data distribution, some truthful and some not. TCH offers a very simple way to quantify the persona hypothesis and provably characterize its influence. Under the TCH, inferring a latent truth variable is loss-reducing: if the model recognizes that "It's well known that the moon landing was a *hoax*" is false, it can raise the probability of a continuation such as "and that the Earth is *flat*," which is likewise false.

We test the truth-co-occurrence hypothesis (TCH) in the *minimal* transformer-like model, with a single self-attention layer, one head, a normalization layer, and no MLP. Training examples are four-token sequences $x\ y\ x'\ y'$ with subjects $x, x'$ ("The capital city of France"; "Churchill's nationality") and attributes $y, y'$ ("Paris"; "British"); with probability $\rho$, the attributes $y, y'$ are *both* the correct attribute; with probability $1 - \rho$, they are replaced with a random one. Under our simplified generative story, "truth" is identified with the attribute that is *frequent* in the training data for a particular subject. When we train LMs on such dataset, we find that after the key–value lookup circuit forms, gradient descent pushes hidden states toward a linear separator that clusters true vs. false contexts, and the model uses it modify its confidence when predicting the attribute. Training shows two phases: rapid key–value acquisition followed by slower emergence of linear encoding. Although our toy model is far simpler than natural training data (see Section 6), it predicts the observed sensitivity to false context (Section 5.3), where false prefixes bias later predictions (supporting TCH), and reproduces the way normalization layers regulates confidence [Stolfo et al., 2024]. Taken together, we show that linear truth encoding can arise without any built-in semantics.

## 2   Related work

A growing body of work shows that pretrained LMs linearly encode a simple notion of "truth"—-consistency with the majority of examples in the training data—in both hidden states and individual MLP/attention outputs [Azaria and Mitchell, 2023, Burns et al., 2022, Li et al., 2024b, Bürger et al., 2025]. This feature is generally robust for frequent atomic facts, though its subspace can shift in the presence of negation [Marks and Tegmark, 2024] and may by biased to dataset-specific features [Orgad et al., 2025]. The encoded truth dimension is behaviorally relevant: intervening on it nudges the model toward truthful completions [Li et al., 2024b] although the model's predictions sometimes do not agree with the latent encoding [Liu et al., 2023]. Yet the *mechanism* behind this encoding remains unclear. Extending the persona hypothesis of Li et al. [2023a], Joshi et al. [2024], Ghandeharioun et al. [2024] link truthful behavior to lexical "personas"—for instance, the formal, encyclopedic style typical of Wikipedia versus the more casual tone common in social-media post. We show that, given sufficient training, LMs also acquire a lexicon-independent abstract truth dimension that emerges more slowly.

The line of work on truth encoding is closely related to findings suggesting that models encode different aspects related to their knowledge and confidence. It was shown that it is possible to decode "latent" knowledge from the model Gekhman et al. [2025], and that measures of uncertainty can be decoded from hidden states [Slobodkin et al., 2023, Farquhar et al., 2024, Ferrando et al., 2025]. Our work is related to, but distinct form, works on mechanistic understanding of hallucinations [Yu et al., 2024]; while both rely on the associative memory used by the model [Geva et al., 2021, 2022a,b, Bietti et al., 2023, Cabannes et al., 2024a], we focus on the emergence of separation between true and false assertions, and come up with a toy model that allows us to analyze its properties.

## 3   The Truth Co-occurrence Hypothesis

We previously described the TCH, the assertion that false statements tend to co-occur. To quantify that, we use the MAVEN-FACT corpus [Li et al., 2024a], where annotators assign a FactBank-style factuality label to *every event mention inside a news article*. After discarding all but **certain** judgments, each mention is labeled CERTAIN-TRUE or CERTAIN-FALSE and grouped by the document

in which it appears.[1] We find the following: *(i)* the overall certain-false rate is $p = 0.0209$; *(ii)* the chance that two event mentions from the *same* article are both certain-false is 0.0009, exceeding the independence baseline $p^2 = 0.00044$ by a factor of $\approx 2$; and *(iii)* the clustering ratio— $\mathrm{Var}_{\mathrm{obs}}(\hat{p}_i)/\mathrm{Var}_{\mathrm{binom}} = 1.23$— shows 23 % extra article-to-article heterogeneity. A $\chi^2$ test of independence confirms the association ($\chi^2 = 4.17 \times 10^3$, $p \approx 9 \times 10^{-49}$). This shows that false assertions are not sprinkled at random but tend to *cluster* on the same article. For a language model, tracking a latent truth bit is therefore loss-reducing: once a page provides evidence that one statement is refuted, the conditional probability that a subsequent claim is also refuted increases. This motivates the design of a simple data-generating process that instantiates the hypothesis and tests whether it gives rise to truth encoding.

## 3.1 Data Generating Process

Natural text confounds *truth* with stylistic cues, topic priors, and corpus frequency [Orgad et al., 2025]. Therefore, Consequently, if we probe LMs on raw text, we risk discovering features that merely track these proxies. To uncover minimal conditions that *force* an LM to represent truth, we build a toy world in which:

1. Every *subject* pair has exactly one canonical attribute (ground truth).

2. A small, controllable fraction of examples are corrupted by uniform noise (the attribute is replaced with another attribute).

3. importantly, the truthfulness of neighboring sequences *correlates*; this models the tendency of speakers to consistently be less or more truthful [Joshi et al., 2024].

Despite its simplicity, this environment reproduces the linear-separability we see in large-scale LMs (§5).

**Data format.** Each training example is a sequence $x \; y \; x' \; y'$ with subjects $x, x' \in \mathcal{S}$ and attributes $y, y' \in \mathcal{A}$.

For every $x$ there exists a unique ground-truth attribute $g(x)$ memorized by the data generator. Examples are corrupted as follows: Sample $T \sim \mathrm{Bernoulli}(\rho)$ once per example, such that

TRUE If $T=1$, set $y_i = g(x), y_i' = g(x')$.

FALSE If $T=0$, draw each $y, y'$ independently and uniformly from $\mathcal{A}$.

**Truth as a latent variable.** Because predicting the *second* attribute token $y'$ is *easier* when $T$ is known, an LM can lower its language-model loss by internally inferring $T$ early in the sequence and propagating that bit forward.

Without inferring $T$, the conditional distribution over the *second* attribute $y'$ given the prefix $(x, y, x')$ is:

$$\Pr\bigl(y' = g(x') \,\big|\, x, y, x'\bigr) = \rho + \frac{1 - \rho}{|\mathcal{A}|}, \qquad \Pr\bigl(y' = a \neq g(x') \,\big|\, x, y, x'\bigr) = \frac{1 - \rho}{|\mathcal{A}|}.$$

Assume the LM can memorize $g$ and (optionally) infer $T$ perfectly. Let $\mathcal{L}_{\neg T}$ be its per-token cross-entropy for predicting $y'$ when it *does not access* $T$, and let $\mathcal{L}_T$ be the loss when it embeds $T$ internally. Then, in the $|\mathcal{A}| \to \infty$ limit, $\mathcal{L}_{\neg T} - \mathcal{L}_T = H_2(\rho)$, the binary entropy of $\rho$. Hence representing a *single bit* yields maximal benefit at $\rho = 0.5$, where $H_2$ is largest (see appendix B for a complete derivation).

## 4 Analysis on a Toy Model

In this section, we study the emergence of truth directions in a simplified one-layer setup with orthogonal embeddings. Empirically, we find that this minimal setup already captures the mechanism of a truth direction, and leverages layer-norm to adjust confidence for the second attribute depending on truthfulness of the first one. Our empirical and theoretical analysis shows that this happens in phases, and that layer-norm is crucial to provide the relevant structure in the gradients. Furthermore,

---

[1]Data-handling details are deferred to App. A.

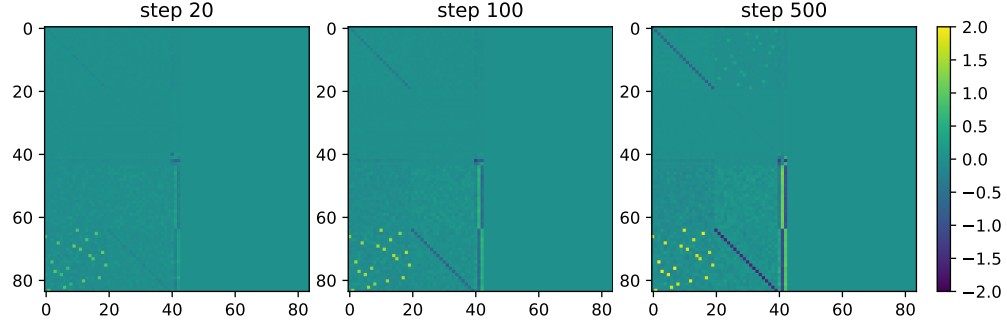

Figure 1: Visualization of the value matrix for the one-layer model at different training steps. We see that the $e_x \to u_{g(x)}$ block is learned first, along with the $p_t \to \bar{u}$ block. Later the $e_x \to -e_x$ and $e_y \to -u_y$ blocks, and finally the $e_y \to e_{g^{-1}(y)}$ block.

such a truth direction can already emerge when there are only true sequences. In appendix D.1, we discuss how these results may be extended to non-orthogonal and to learned embeddings.

**Setup.** Consider the following one-hot token embedding, positional embedding, and unembedding vectors in $\mathbb{R}^d$ with embedding dimension $d = 4N + 3$, where $z \in [2N]$ is an input or output token (input tokens $x$ are in $[N]$ while outputs $y$ are in $[N+1, 2N]$), and $t \in [3]$ a position:

$$[e_z]_i = \mathbf{1}\{i = z\} \tag{1}$$
$$[p_t]_i = \mathbf{1}\{i = 2N + t\} \tag{2}$$
$$[u_z]_i = \mathbf{1}\{i = 2N + 3 + z\}. \tag{3}$$

We consider a one-layer transformer with uniform causal attention, and a basic layer-norm operation. Concretely, for an input sequence $z_{1:3} = (x, y, x')$ and position $t \in [3]$, define:

$$F_W(z_{1:t})_t = U \cdot \mathsf{N}\left(e_{z_t} + p_t + \frac{1}{t}\sum_{s=1}^{t} W(e_{z_s} + p_s)\right), \tag{4}$$

where $W$ denotes the value matrix, $U = u_{1:2N}^{\top} = [\mathbf{0}; I_{2N}] \in \mathbb{R}^{2N \times d}$ is a projection on the unembedding dimensions, and $\mathsf{N}(v) = v/\|v\|$ is a layer-norm operation. The predicted probabilities are then given by $\hat{p}(z_{t+1} = \cdot | z_{1:t}) = \mathcal{S}_\beta(F(z_{1:t}))$, where $\mathcal{S}_\beta$ denotes the softmax operation with inverse temperature $\beta$. Our experiments use $\beta = \sqrt{d}$, due to the use of RMS norm in layer-norm over embeddings of dimension $d$.

We assume here that $x, x' \sim \text{Unif}([N])$ i.i.d., and conditioned on these as well as on a truth random variable $T \sim \text{Ber}(\rho)$, we have $y = g(x)$ and $y' = g(x')$ when $T = 1$, and $y, y' \sim \text{Unif}([N+1, 2N])$ otherwise. Denoting $z_{1:4} = (x, y, x', y')$, the population loss then takes the form

$$L(W) = \sum_{t=1}^{3} L_i(W) = \sum_{t=1}^{3} \mathbb{E}_{z_{1:t+1}}\left[-\log \mathcal{S}_\beta(F_W(z_{1:t}))_{z_{t+1}}\right]. \tag{5}$$

**Probing the mechanism and its emergence.** Figure 1 shows a visualization of the value matrix $W$ in our toy model, at different steps of training, with $N = 20$, $\rho = 0.8$ and batch size 16. We see that a clear block-structure emerges in the matrix $W$, with different blocks arising in different phases. Some blocks show a negative identity structure, while others show a permutation structure according to the "knowledge" mapping $g$. Positional embeddings show more uniform patterns across unembeddings, with different signs depending on whether the next token is an input or label. In Figure 2, we show the representations at the $x'$ token for examples of true and false sequences, before and after layernorm, as well as the probabilities obtained after projecting to the unembedding space and applying softmax. In the false sequence (bottom plot), we notice large spikes in the input embedding dimensions (1-20) at positions $x = 5$ and $g^{-1}(y) = 16$. These do not exist in true sequences, since they cancel out. We see a similar behavior on unembedding dimensions (65-84) at smaller scales. The cancellation leads to a smaller norm on true sequences, which causes an amplification of the logits, and finally a spiked distribution on true sequences, versus a flatter one on false sequences, though we still some lower confidence spikes on $g(x)$ and $g(x')$ (note the $y$-axis scale difference).

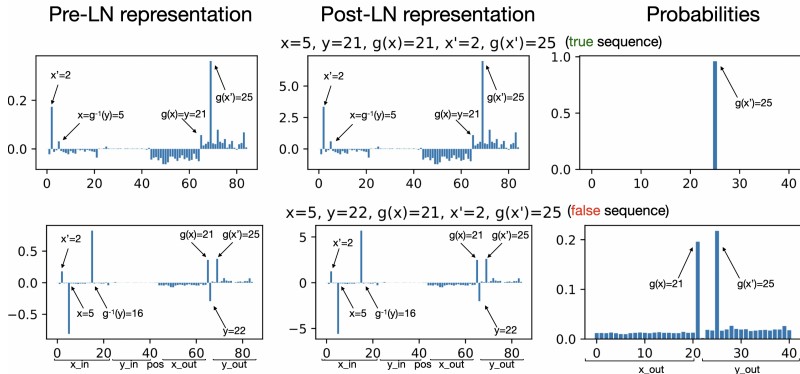

Figure 2: Visualization of representations on true (top) and false (bottom) sequences. The plots show representations before (left) and after (center) layer-norm, as well as predicted probabilities (right).

**Structure of the value matrix $W$.**  We now study a construction that resembles the one observed empirically in fig. 1. Later we will provide a theoretical justification for this structure and its emergence in phases by analyzing training dynamics.

The leftmost column of the $W$ matrix maps $e_x$ to its corresponding label $u_{g(x)}$, while also subtracting $e_x$ itself:

$$W e_x = -\alpha_1 e_x + \beta_1 u_{g(x)}, \tag{6}$$

with $\alpha_1, \beta_1 > 0$. The second column has the following symmetric behavior:

$$W e_y = \alpha_2 e_{g^{-1}(y)} - \beta_2 u_y. \tag{7}$$

Finally, the third column maps the different positional embeddings to mixtures of uniform distributions over the inputs or labels:

$$W p_1 = \gamma_1 (\sum_y u_y - \sum_x u_x) \tag{8}$$

$$W p_2 = -\gamma_2 (\sum_y u_y - \sum_x u_x) \tag{9}$$

$$W p_3 = \gamma_3 (\sum_y u_y - \sum_x u_x). \tag{10}$$

In the statements above, we assume all the coefficients $\alpha_{1/2}, \beta_{1/2}, \gamma_{1/2/3}$ to be positive.

**Linear separation and sharpening mechanism.**  One important consequence of the structure above is that any token that attends to both $x$ and $y$ (this could be either $y$ or $x'$) has the following quantity in its residual stream:

$$\zeta(x, y) := W(e_x + e_y) = -\alpha_1 e_x + \alpha_2 e_{g^{-1}(y)} + \beta_1 u_{g(x)} - \beta_2 u_y. \tag{11}$$

We then have

$$\|\zeta(x, g(x))\|^2 = \|\zeta(x, y)\|^2 - 2\alpha_1 \alpha_2 - 2\beta_1 \beta_2, \quad \text{for } y \neq g(x).$$

Since $\alpha_1, \alpha_2, \beta_1, \beta_2 > 0$, the norm of $\zeta$ on a true sequence is always smaller than on a false sequence, leading to a useful feature for detecting truth (see illustration in fig. 16). Combined with the layer-norm operation, this provides a mechanism for *sharpening* the prediction of $y'$ towards $g(x')$ when the model detects a true sentence, by adjusting the temperature in the softmax via inverse norm scaling.

**Theorem 1** (Sharpening of $y'$ predictions). *Suppose we have a solution that satisfies Eqs.* (6)-(10). *Denote by $c := 2 + \frac{\bar{\gamma}^2(2N-2) + 2\alpha_1^2 + \beta_1^2}{9}$ For any $x, x'$ and $y \neq g(x)$, we have:*

$$F(x, g(x), x')_{g(x')} - \max_{k \neq g(x')} F(x, g(x), x')_k \geq \frac{\beta_1 - \max(0, \beta_1 - \beta_2)}{3\sqrt{c + (\beta_1 - \beta_2 + \bar{\gamma})^2 + (\beta_1 + \bar{\gamma})^2}}$$

$$F(x, y, x')_{g(x')} - \max_{k \neq g(x')} F(x, y, x')_k = 0$$

The proof is in appendix E.2. This shows that the structure of $W$ along with layer-norm provide a simple mechanism to make the model more confident about its knowledge when the context is truthful. For false sequences, the zero gap comes from the fact that logits for $g(x)$ and $g(x')$ are tied, as we show empirically in Figure 2. This aligns with previous interpretability work on confidence neurons [Stolfo et al., 2024]. Beyond improving prediction performance, we now show that this model provides a linear encoding of truth in the representations after layer-norm.

**Theorem 2** (Linear truth direction). *Suppose we train the model in* (4) *as explained above, and reach a solution for $W$ that satisfies Eqs.* (6)-(10). *Then, we have the following:*

1. *If the model in* (4) *does not contain* N, *then its output on the $y$ token does not admit a linear separator for true and false samples.*

2. *If the model in* (4) *contains* N *and* $2\alpha_1\alpha_2 + 2\beta_1\beta_2 \neq 0$, *then its output on the $y$ token admits a linear separation for true and false samples. Moreover, if $\gamma_1 = \gamma_2$, $\alpha_1 = \alpha_2$, $\beta_1 = \beta_2$ then the margin is at least $\delta = \frac{1}{2\sqrt{2}}\left(1 - \frac{1}{\sqrt{1+\alpha^2+\beta^2}}\right)$.*

The proof is in appendix E.3.

**Theoretical analysis of training dynamics.**  We now study how such a structure in $W$ emerges from training dynamics in a simplified setting.

**Theorem 3** (Sequential gradient learning; informal). *In a simplified model with no positional embeddings, taking two gradient steps on $L_1$ (predicting $y$ from $x$) followed by one on $L_3$ (predicting $y'$ from $x'$), all with step-size $\Theta(N)$, leads to the desired structure for $W$ as in Eqs.* (6)-(7), *up to negligible entry-wise $O(1/N)$ terms.*

See a formal statement and a proof in appendix E.1. This result shows that gradient dynamics in our model can quickly lead to the block structure observed in Figure 1, despite the non-convexity induced by normalization. In fact, the analysis reveals that the layer-norm operation is crucial here to obtain many of the desired blocks other than the $e_x \to u_{g(x)}$. Interestingly, our theory shows that this structure arises even when $\rho = 1$, and empirically we found that both sharpening and linear separation indeed happen in this setting, demonstrating an emergent out-of-distribution generalization to false sequences. We note, however, that this may not happen in a more expressive model: we empirically found that if we also train the key-query matrix with $\rho = 1$, the model quickly learns to focus its attention to the current token (see also appendix E.1.1 for a theoretical justification), which makes information from the context inaccessible from the residual stream. While this may improve predictions of $g(x')$ on true sequences by removing noise in the residual stream coming from $(x, y)$, this also results in a failure to handle false sequences.

## 5 Experiments

### 5.1 Synthetic Setting

**Setup.**  We train minimal self-attention LMs on the synthetic dataset described above. The model contains $l$ self-attention layers with a single attention head, followed by layer normalization, with no feedforward network. See Appendix C for more details. In this setting, in contrast to the toy model described above, we train all parameters, including the dense embeddings and the attention module.[2] Each training example is a concatenation of a subject ($x$), an attribute ($y$), an additional, uniformly-sampled subject ($x'$), and an additional attribute ($y'$). The attributes $y, y'$ are either sampled uniformly or taken to be the correct attributes $g(x), g(x')$, according to the true probability $\rho$. In line with the Truth Co-occurrence Hypothesis, we aim to measure whether in this training setting the model is able to recover the latent truthfulness of the first sequence (verifying whether $y = g(x)$) and *use* it to decrease LM loss on the second attribute $y'$.

We experiment with true-attributes rates $\rho$, and with $l \in \{1, 2, 3\}$ layers, and assume a perfect correlation between the truthfulness of the first and second attributes (that is, $y = g(X)$ if, and only if, $y' = g(x')$). Along training, we fit logistic-regression classifiers on all hidden states to predict

---

[2]We release the code in https://github.com/shauli-ravfogel/truth-encoding-neurips.

whether or not the sequence is false (a binary classification problem). We fit individual classifiers both the first attribute position ($y$), as well as on the second subject position ($x'$), from which the second attribute $y'$ is predicted. While in training the LM we use a varying true-attribute rate $\rho$, the linear classifiers are always trained and evaluated on a *balanced* set, containing 50% true sequences. We report mean results over 5 runs with different random seeds. Unless specified otherwise, we present here results for $l = 1$ and $\rho = 0.99$, $|\mathcal{A}| = |\mathcal{S}| = 512$ and $d_{\text{model}} = 256$; results for other settings are deferred to Appendix D.

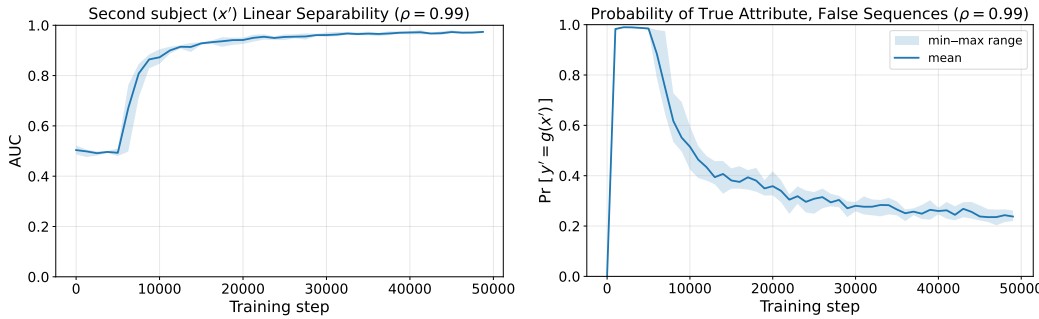

(a) Truth classification results, second subject $x$    (b) $P[g(x')]$ on *false* sequences, for which $y' \neq g(x')$)

Figure 3: Truth linear classification results alongside probability assigned by the LM to the true attribute on *false* sequences.

## 5.2 Results

**Two-phase dynamics.** In Figure 3a we show the linear truthfulness classification AUC as a function of training steps, on the second subject. for a $1-$layer model with true-attribute probability $\rho = 0.99$. Additionally, we plot the probability the LM assigned to the *correct* attribute on *false* sequences ($P(y' = g(x') \mid y \neq g(x))$; Figure 5b). When this probability is minimized, the model improves its loss on false sequences.

In line with the toy model, we detect distinct phases in training.

1. **Memorization.** As can be seen in Figure 5b, memorization happens rapidly—within the first 1000 batches—as the model converges to a probability of around 1 to $g(x')$ on *both* true and false examples. Indeed, the model predicts the correct attributes on over 99% of the true sequences.

2. **Truth encoding.** The model does learn to linearly encode the truth latent variable. This encoding emerges abruptly, after around 7,500 batches, during which the model saw around 1 million examples, relatively long after the model achieves perfect memorization.

The model learns to decrease the probability it assigns to the correct attribute on the second attribute position $P(g(x') = y')$ roughly at the same time linear classification emerges.

**Truth circuit.** We aim to understand how the linear truth subspace is being computed. While it has been empirically shown that LM linearly encode many human-interpretable concepts [Bolukbasi et al., 2016, Vargas and Cotterell, 2020, Ravfogel et al., 2022], it is not well-understood *why* linear representations emerge in hidden layers [Park et al., 2024, Jiang et al., 2024]. The toy model we propose allows us to empirically study the origins of the linear signal, and the way it is being used to decrease LM loss on the second attribute.

The truth encoding appears in a 1-layer model (classification accuracy in the input embeddings layer is at majority level). As can be seen in the first layer attention pattern in (Figure 4b), this attention head calculates an approximate mean of the embeddings of $x$ and $y$, after application of the $V, O$ self attention matrices, in line with the uniform attention assumed in the toy model. One key difference is that here, we learn the input embeddings. Interestingly, inspecting the PCA of the input subject and attribute tokens (Figure 4c) reveals that approximately, $e_x = -e_{g(x)}$ on the first principal component. This explains why both the true and false representations tend to cluster around the origin. Following the attention averaging, we apply RMSNorm. We find that linear classification emerges only *after* normalization; classification accuracy is at majority level before it. Indeed, a PCA plot (Figure 4)

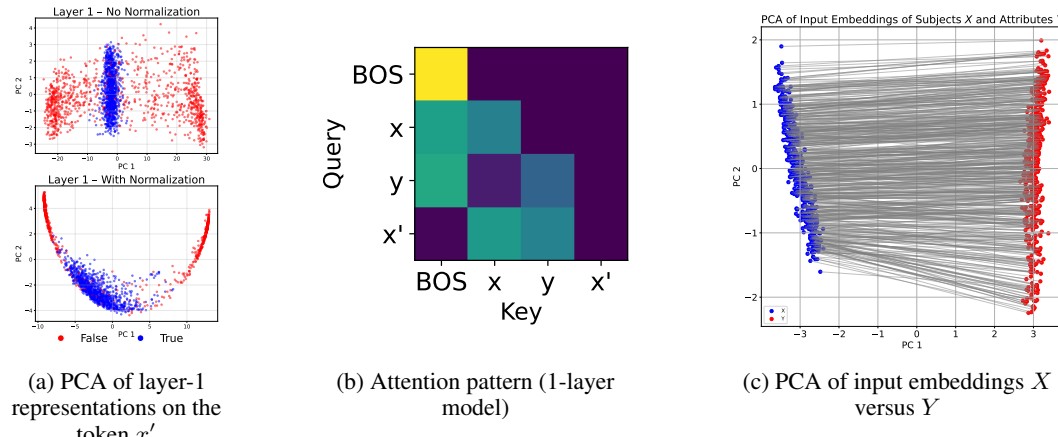

(a) PCA of layer-1 representations on the token $x'$

(b) Attention pattern (1-layer model)

(c) PCA of input embeddings $X$ versus $Y$

Figure 4: LM representations over *false* sequences.

shows that, as predicted by the toy model, the TRUE class is centered around the origin, with a larger variance for the TRUE class than the FALSE class. Normalization induces linear separability, that is also evident in the first 2 PCA components.

**Additional settings.** So far we have analyzed a single-layer self-attention network—either with one-hot embeddings, or with trainable dense embeddings and $\rho = 0.99$. Results for other configurations appear in Appendix D; here we outline the main trends. The patterns in Figures 3a and 5b persist across layer counts $l$, noise levels $\rho$, and corpus sizes $|\mathcal{S}|, |\mathcal{A}|$. Higher $\rho$ delays (but does not prevent) the onset of linear separability, which still emerges at $\rho = 0.999$ (Figure 7a at the appendix); only the degenerate case $\rho = 1.0$ shows no emergence, contrary to the toy model. We discover similar structures to Figure 1 also when training with frozen dense embeddings and when learning the $KV$ matrices instead of using fixed attention. A preliminary analysis of this setting is provided in appendix D.1 and appendix E.1.1, and we leave a more complete understanding for future work. With additional layers the model sometimes encodes truth in the first attribute $y$, then *copies* it to $x'$ before predicting $y'$; in other runs it reverts to the single-layer strategy where $x'$ attends directly to $x$ and $y'$. This influences whether we see linear encoding on *both* $y$ and $x'$, or on $x'$ alone (Figure 7b in the appendix).

## 5.3 Testing the TCH in a Real LM

The theory we specified relies on a set of assumptions and architecture that do not exist in pretrained transformers (those have, for instance, MLP layers in addition to the attention layer; have multiple attention heads; and are trained primarily on natural language distributions). Below, we (i) train "regular" transformer models on a *natural language* data that instantiates the truth co-occurrence hypothesis; (ii) assess to what extent aspects of the mechanism we propose exist in pretrained LLMs.

### 5.3.1 Instantiating the TCH in Natural Language

In section 5.1, we created a synthetic dataset that respects the TCH and showed that training an attention-only transformer on this data results in linear truth encoding. Here, we aim to assess whether the same thing happens when training "real" transformers on natural language data.

**Setup.** We evaluate on the CounterFact dataset [Meng et al., 2022], a collection of simple factual assertions spanning relations such as SPEAKSLANGUAGE and BORNIN. We select the 25 most frequent relations and, for each positive instance $(x, r, a)$, construct a negative by replacing the attribute $a$ with a different attribute from the same relation. To instantiate the TCH, we form paired examples by concatenating two randomly sampled instances that share the same truth label (both true or both false). We then train a small transformer with RMS normalization, 2 attention heads and a single MLP module per layer, hidden size $d = 256$, and depth $l \in \{2, 5, 9\}$ on this corpus. We use

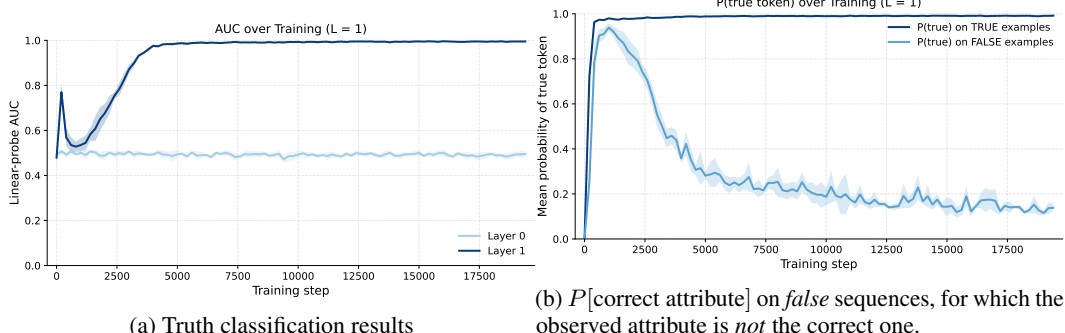

(a) Truth classification results

(b) $P[\text{correct attribute}]$ on *false* sequences, for which the observed attribute is *not* the correct one.

Figure 5: Truth linear classification results alongside probability assigned by the LM to the true attribute on *false* sequences.

$\rho = 0.99$. We train on data from a single relation at a time, and report mean and standard deviations over 5 random relations.[3]

**Results.** Across all seeds and architectural choices, the training dynamics mirror those on synthetic data: rapid memorization, followed by the emergence of a linear encoding, and an increase in entropy on false sequences. In fig. 5, we show results for a single relation (WORKSIN; averaged over five random seeds). By the end of training, the final hidden layer is nearly perfectly separable by the truth label, and on *false* sequences the probability assigned to the memorized ("true") attribute declines. Notably, the 1-layer model exhibits *epoch-wise double descent*: classification accuracy rises early, dips, and then rises again. Across the five seeds, relations, and model sizes, memorization proceeds at roughly the same rate; the main variance lies in how quickly the probability declines on false sequences.

### 5.3.2 The TCH in Pretrained LLMs

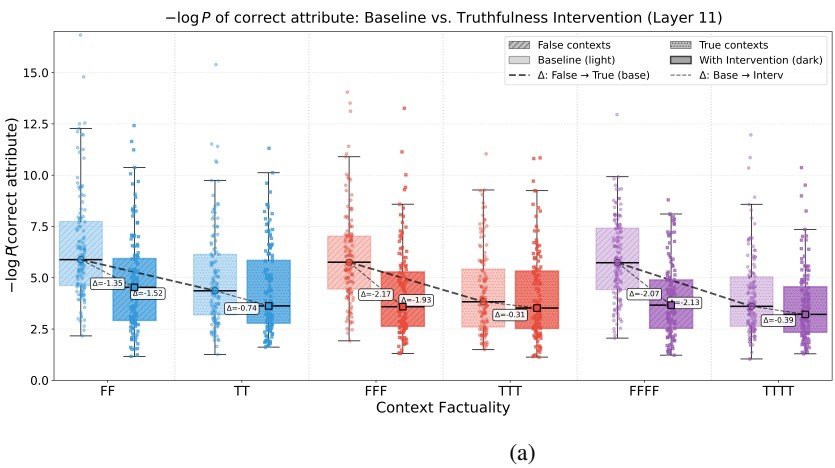

(a)

The mechanism proposed above assumes a very specific data generating process, and a simplified transformer model. As such, it is not likely that the same mechanism applies to real LMs; we see the toy model as a proof of concept, and aim to study more complicated models in future work. Yet, in this section, we compare the predictions following from our hypothesis with pretrained LMs in these aspects: (1) the sensitivity of the model's predictions to preceding false sentences, in line with the truth co-occurrence hypothesis; (2) the behavioral relevance of the linear truth encoding in a situation where a sentence follows misleading false sentences. We experiment with a LLAMA3-8B model [Grattafiori et al., 2024] and the CounterFact dataset (SPEAKSLANGUAGE relation). We let the model predict the first token of the last word of a sentence, when it is (i) preceded by $n$ false

---

[3]We leave the question of *generalization* between relations to a future work.

sentences; or (ii) preceded by $n$ true sentences. In line of the hypothesis, we expect to see a decrease in the probability of the correct answer.

**Model's predictions are sensitive to preceding false sentences.** The results, over 128 $n-$tuples, are presented in Figure 6a (light bars) and are in line with our hypothesis; for instance, in the two leftmost box plots, we see that preceding the sentence with two false sentences (*FF*) yields higher negative likelihood (smaller probability) to the correct attribute compared with when preceding it with one true sentence (*TT*). The difference in negative log likelihood is $1.52$, corresponding to $4.55\times$ decrease in the probability of the correct attribute.

**Intervention in the truth subspace.** LLAMA3-8B encodes truthfulness linearly: a linear classifier reaches over 95% accuracy on all middle and last layers in separating true instances from the dataset from counterfactual ones. Our theory predicts that, in the presence of misleading context, the direction that distinguishes true from false vectors actively pulls the model away from the correct answer. To test that, we intervene in the truth subspace. Following previous work on linear steering [Li et al., 2023b, Singh et al., 2024], we calculate the mean vector of the TRUE and FALSE classes in the representation space, $\mu_T$ and $\mu_F$, and add a steering vector $\alpha(\mu_T - \mu_F)$ to all representations in the same layer with the goal of increasing the probability of the correct attribute. We choose layer $l = 11$ based on preliminary experiments that showed that classification peaks at that layer, and $\alpha = 3.0$. The results, presented in Figure 6a (darker bars), show that the models tend to increase the probability of the correct attribute post-intervention, even in the presence of false context.

**Emergence along training** See appendix E.4 for a preliminary analysis of the emergence of truth encoding along training.

## 6 Discussion and Limitations

Although our analysis was grounded in a deliberately minimalist transformer, it discovers a two–phase dynamic—rapid key–value memorization followed by the slower emergence of a *linear truth encoding*. The key prerequisite appears to be the presence of (i) an associative–memory circuit able to retrieve subject–attribute pairs and (ii) correlation among the truth values of adjacent clauses. We emphasize that this is *one*, and probably not a unique, mechanism that can induce truth encoding. A core advantage of the minimalist model is that it does not assume any lexical cues that help the model discern the truth latent variable. In that sense, this is a more challenging setting than the previously studied one [Joshi et al., 2024], where it is assumed that true and false assertions are associated with different lexical distributions.

Several core differences exist between our simplified generative story and a real-world setting. Our synthetic corpus contains only one latent relation. A natural extension is to sample tuples from a set of heterogeneous relations—BORNIN, CAPITALOF, CURRENCYOF, . . . —while maintaining correlation in the *latent* truth bit. Doing so forces the model to *contextualize* its memory: the same subject embedding must participate in multiple key–value slots distinguished by the relation. Real corpora have *logical* and *semantic* dependencies that go far beyond pairwise subject–attribute pairs: transitivity (*"A is in B"* $\wedge$ *"B is in C"* $\Rightarrow$ *"A is in C"*), mutual exclusivity (*"isAlive"* vs. *"IsDead"*), and type constraints (*"capitalOf"* only applies to geopolitical entities). These constraints also greatly limit the range of plausible counterfactual variants we may see in the training data; while we assume a uniform corruption for simplicity, in practice false variants of factual claims come from a unique conditional distribution.

## 7 Conclusion

We introduced a small transformer and a synthetic data-generation process that jointly suffice to yield a robust *linear truth subspace*. Our analytical and empirical results demonstrate a two-phase training dynamic: memorization followed by truth-code emergence. Unlike prior persona-based accounts, our theory does not rely on surface correlations between individual tokens and truthfulness, and points out to a possible mechanism behind the emergence of linear of the truth signal as a *latent variable* inferred by the model.

## Acknowledgments

This work was supported in part through the NYU IT High Performance Computing resources, services, and staff expertise, and by the National Science Foundation (NSF) under Grant No. IIS-2239862 to TL. We thank Yanai Elazar for his valuable comments on a previous version of this paper.

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

# Appendix

## A Detailed MAVEN-FACT Analysis

**Data extraction.** We use the *train* split of MAVEN-FACT v1.0 (73,939 event–mentions drawn from 2 913 news articles).[4] Each mention carries a FactBank-style factuality code (CT++, CT+, CT-, CT-, PS$\pm$, PR$\pm$, CF$\pm$, U, NA, ...). We retain only **certain** judgments:

$$\text{certain-true} = \{\text{CT++}, \text{CT+}\}, \qquad \text{certain-false} = \{\text{CT- -}, \text{CT-}\}.$$

All other codes are discarded, leaving $N = 71,274$ labelled mentions.

**Grouping key.** Mentions are grouped by their originating article ID (`doc_id`), giving $M = 2,913$ documents with at least two certain mentions ($n_i > 1$). Let $Z_{ij} \in \{0, 1\}$ indicate whether mention $j$ in document $i$ is *certain-false*.

**Statistics reported in the main text.**

- **Corpus certain-false rate.** $p = \frac{1}{N} \sum_{i,j} Z_{ij} = 0.0209$.

- **Pairwise certain-false probability.** $\Pr(Z_j = Z_k = 1 \mid \text{same doc}) = \frac{\sum_i \binom{f_i}{2}}{\sum_i \binom{n_i}{2}} = 0.00090$, where $f_i = \sum_j Z_{ij}$.

- **Independence baseline.** $p^2 = 0.00044$.

- **Clustering ratio.** $\frac{\text{Var}_{\text{obs}}(\hat{p}_i)}{\text{Var}_{\text{binom}}} = \frac{\frac{1}{M} \sum_i (\hat{p}_i - p)^2}{\frac{1}{M} \sum_i p(1-p)/n_i} = 1.23$, with $\hat{p}_i = f_i/n_i$.

- **$\chi^2$ test.** The $2 \times M$ contingency table of $\{f_i, n_i - f_i\}$ yields $\chi^2 = 4\,174$ ($p \approx 9 \times 10^{-49}$).

These figures show that *certain-false* events, though rare (2.1%), occur about twice as often as chance would predict when two events come from the same article, and the distribution of false rates across articles is 23 % more heterogeneous than a binomial model would permit—confirming the co-occurrence signal predicted by TCH.

The MAVEN-ED dataset is released with CC BY-SA 4.0 license. The MAVEN-ARG and MAVEN-ERE are published with GPLv3 license.

## B Entropy incentive

**Setup.** We consider sequences $(x, y, x', y')$ with subjects $x, x' \in \mathcal{S}$ and attributes $y, y' \in \mathcal{A}$. Let $g : \mathcal{S} \to \mathcal{A}$ be the ground-truth attribute map. A latent bit $T \sim \text{Bernoulli}(\rho)$ governs whether attributes are truthful ($T{=}1$) or random ($T{=}0$):

$$T = 1 : \ y = g(x), \ y' = g(x') \quad \text{(deterministic)}; \qquad T = 0 : \ y, y' \overset{\text{i.i.d.}}{\sim} \text{Unif}(\mathcal{A}).$$

We study the optimal next-token loss for predicting $y'$ given the prefix $(x, y, x')$ under two cases: (i) the model *does not access* $T$; (ii) the model *does* access $T$ (and can memorize $g$).

**A. Predictive distribution of $y'$ without access to $T$**

By the law of total probability over $T$ and the generator above,

$$\Pr\big(y' = g(x') \,\big|\, x, y, x'\big) = \Pr(T{=}1 \,|\, x, y, x') \cdot 1 \ + \ \Pr(T{=}0 \,|\, x, y, x') \cdot \tfrac{1}{|\mathcal{A}|}. \tag{12}$$

When we say the model is "ignorant of $T$," we mean it does not exploit any posterior signal about $T$ from the prefix; thus we use the prior $\Pr(T{=}1 \,|\, x, y, x') = \rho$ and $\Pr(T{=}0 \,|\, x, y, x') = 1 - \rho$. Hence

$$\Pr\big(y' = g(x') \,\big|\, x, y, x'\big) = \rho + \frac{1 - \rho}{|\mathcal{A}|}. \tag{13}$$

For any *specific wrong* $a \in \mathcal{A} \setminus \{g(x')\}$,

$$\Pr\big(y' = a \,\big|\, x, y, x'\big) = \rho \cdot 0 + (1 - \rho) \cdot \frac{1}{|\mathcal{A}|} = \frac{1 - \rho}{|\mathcal{A}|}. \tag{14}$$

---

[4]Available at `https://github.com/THU-KEG/MAVEN-FACT`.

**B. Optimal per-token cross-entropy without $T$**

Let $\alpha := \rho + \frac{1-\rho}{|\mathcal{A}|}$ and $\beta := \frac{1-\rho}{|\mathcal{A}|}$. The optimal model (that does not access $T$) matches the true conditional in (13)–(14), so its per-token cross-entropy equals the entropy of that distribution:

$$\mathcal{L}_{\neg T} = H\Big(\alpha, \underbrace{\beta, \ldots, \beta}_{|\mathcal{A}|-1 \text{ times}} \Big) = -\alpha \log \alpha - (|\mathcal{A}| - 1)\,\beta \log \beta. \tag{15}$$

**C. Optimal per-token cross-entropy with access to $T$**

If the model *does* access $T$ (and has memorized $g$):

$T = 1 :$ loss $0$ (since $y' = g(x')$ deterministically); $\qquad T = 0 :$ loss $\log |\mathcal{A}|$ (uniform on $\mathcal{A}$).

Averaging over $T$ gives

$$\mathcal{L}_T = (1 - \rho) \log |\mathcal{A}|. \tag{16}$$

**D. The gap and its limit as $|\mathcal{A}| \to \infty$**

Subtracting (16) from (15):

$$\Delta := \mathcal{L}_{\neg T} - \mathcal{L}_T = -\alpha \log \alpha - (|\mathcal{A}| - 1)\beta \log \beta \; - \; (1 - \rho) \log |\mathcal{A}|. \tag{17}$$

Using $\beta = \frac{1-\rho}{|\mathcal{A}|}$,

$$-(|\mathcal{A}| - 1)\beta \log \beta = -(1 - \rho)\Big(1 - \tfrac{1}{|\mathcal{A}|}\Big) \log(1 - \rho) + (1 - \rho)\Big(1 - \tfrac{1}{|\mathcal{A}|}\Big) \log |\mathcal{A}|.$$

Plugging into (17) and simplifying,

$$\Delta = -\alpha \log \alpha \; - \; (1 - \rho)\Big(1 - \tfrac{1}{|\mathcal{A}|}\Big) \log(1 - \rho) \; - \; \frac{1 - \rho}{|\mathcal{A}|} \log |\mathcal{A}|. \tag{18}$$

Since $\alpha = \rho + \frac{1-\rho}{|\mathcal{A}|} \to \rho$ and the last term is $O(\frac{\log |\mathcal{A}|}{|\mathcal{A}|})$, we obtain the limit

$$\Delta \xrightarrow[|\mathcal{A}| \to \infty]{} -\rho \log \rho - (1 - \rho) \log(1 - \rho) \; \equiv \; H_2(\rho) \quad \text{(up to the log base).} \tag{19}$$

## C  Experimental Setup

**Model**. We experiment with an attention-only transformer with a single attention head with a post-attention LN:

$$X^0 = E + P \qquad\qquad\qquad\qquad \text{// } E, P \in \mathbb{R}^{V \times d} \text{ (token + positional embeddings)} \tag{20}$$

$$Q^{(i)} = X^{(i-1)} W_Q^{(i)}, K^{(i)} = X^{(i-1)} W_K^{(i)}, V^{(i)} = X^{(i-1)} W_V^{(i)} \tag{21}$$

$$A^{(i)} = \mathrm{softmax}\Big(\frac{Q^{(i)} K^{(i)\top}}{\sqrt{d}}\Big) V^{(i)} \qquad\qquad \text{// attention mix } A^{(i)} \in \mathbb{R}^d \tag{22}$$

$$\tilde{A}^{(i)} = A^{(i)} W_O^{(i)}, \quad W_O^{(i)} \in \mathbb{R}^{d \times d} \qquad\qquad \text{// single-head attention output} \tag{23}$$

$$X^{(i)} = \mathrm{N}\big(X^{(i-1)} + A^{(i)}\big), \qquad i = 1, \ldots, l \qquad \text{// residual + normalization} \tag{24}$$

$$Z = X^{(l)} W_O + b_O, \quad W_O \in \mathbb{R}^{d \times V}, \; b_O \in \mathbb{R}^V \tag{25}$$

$$\hat{Y} = \mathrm{softmax}(Z) \tag{26}$$

**Experiments with one-hot models (section 4)**. The theoretical analysis is driven by experiments on models equipped with frozen, one-hot embeddings and uniform attention, the latter obtained by setting the attention-key matrix $K$ to the zero matrix. Under these conditions the *columns* of the attention value–output product $KV^\top$ map directly to individual vocabulary items, exposing a clear block structure in the matrix (fig. 1). As detailed in the main text, the vocabulary is organized so that indices 1–20 encode input subject embeddings, 21–40 input attribute embeddings, 41–44 positional embeddings, 45–64 output subject embeddings, and 65–84 output attribute embeddings.

**Methodology: interpreting one-hot embeddings.** Figure 2 contrasts two sequences—a correct one (top row) and an incorrect one (bottom row)—by showing the final-layer activations before projecting to the logit space. The one-hot embeddings make the activation patterns in that layer interpretable. We display the activations for the raw representations (left), after layer normalization (middle), and after applying the unembedding matrix and the softmax transformation (right). Observe the differing $y$-axis scales: normalization substantially magnifies the component corresponding to the correct answer in the "true" sequence, while the effect is far less pronounced for the false sequence. The model that produced fig. 1 was trained with SGD, learning rate 1.0 and batch size 16. The output matrix was fixed to identity, and only the value matrix was learned, from zero initialization.

**Experiments with fully-trained models (section 5)**: In section 5, we train all components, including the input embeddings and the $K$ attention matrix. The model is trained for 50,000 batches of size 128 and is optimized with the Adam optimizer [Kingma and Ba, 2015] with a learning weight of 1e-4 and a weight decay of 1e-5. We do not include biases in the attention modules, and use RMSNorm as layer normalization. We run all experiments on 4 NVIDIA GeForce GTX 1080 GPUs. Training a single model lasts up to half an hour.

# D  Additional Experiments

In the main text we concentrated on a single-layer model ($l = 1$) with a true-attribute probability of $\rho = 0.99$. Here we extend the analysis to additional settings.

Our primary focus was the linear separability at the second-subject token, $x'$, where the model predicts the second attribute. This is the only position where the truth signal is *behaviorally* relevant. Nevertheless, the theory also predicts a linear truth encoding at the first-attribute token $y$, owing to the fixed attention pattern. When the attention $KV$ matrix is learned, however, this need not occur—the model can rely exclusively on the attention paid to $x'$ and leave $y$ uninformative. The same theory further implies that a linear truth direction should eventually emerge for any true-sentence rate $\rho$, even though the gradient magnitude (and therefore the speed of emergence) does depend on $\rho$.

**Varying the true sentence rate, $\rho$.** In fig. 7b we vary $\rho$ across five random seeds and measure linear separability at both token positions. As predicted, when the attention pattern is learned, separability is much stronger at the second subject than at the first attribute. The time to emergence grows as $\rho$ increases, yet linear encoding still appears even at the extreme setting of $\rho = 0.999$. Developing a theory that precisely predicts this $\rho$-dependent timing is left to future work.

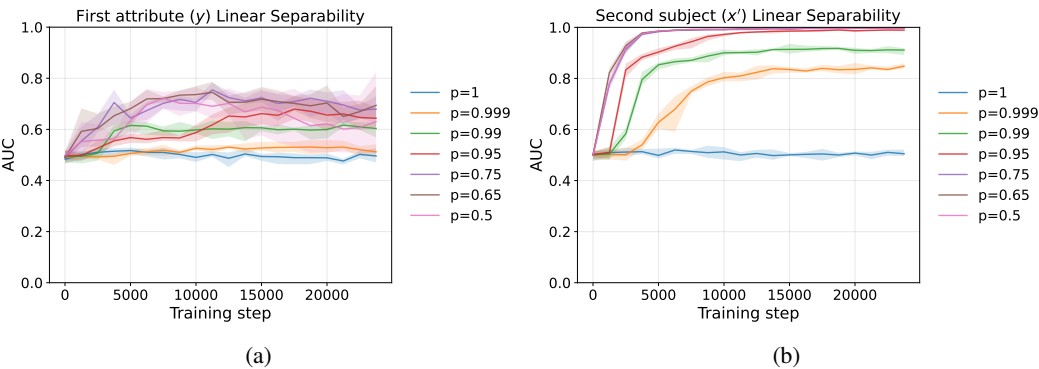

(a)                                                                 (b)

Figure 7: Dependency of linear separability on $\rho$.

**Dependency on $d_{\textbf{model}}$ and $|\mathcal{S}|$.** In fig. 8 we plot the linear separability at the final checkpoint, for different hidden sizes and number of facts to memorize ($\rho = 0.99$, $l = 1$ are fixed). With the exception of $d_{\text{model}} = 32$, the separability persists over the second subject $x'$ for different combinations of these parameters.

**Additional layers**. As we discuss in the main-text (section 5), in a model with a single self-attention layer, it is the second attribute ($x'$) token that attends to both $x$ and $y$. With more layers, there are additional strategies. For instance, $y$ may attend to both $x$ and itself in the first layer, in the same way $x'$ attends to both $x$ and $y$ in the theoretical 1-layer model; then, in the next layer, $x'$ attends to $y$,

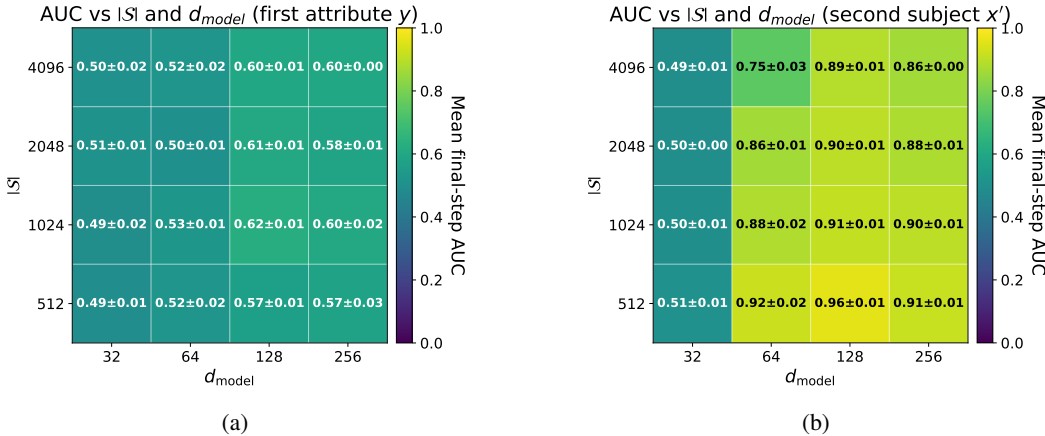

Figure 8: Dependency of linear separability on $d_{\mathrm{model}}$ and $|\mathcal{S}|$.

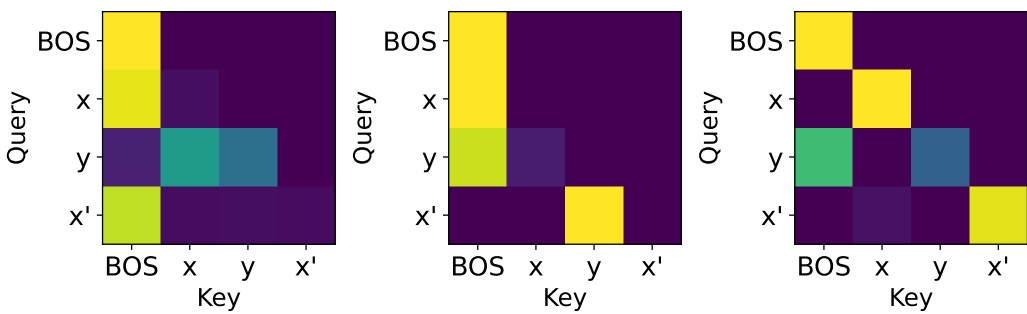

Figure 9: attention patterns of a 3-layer model.

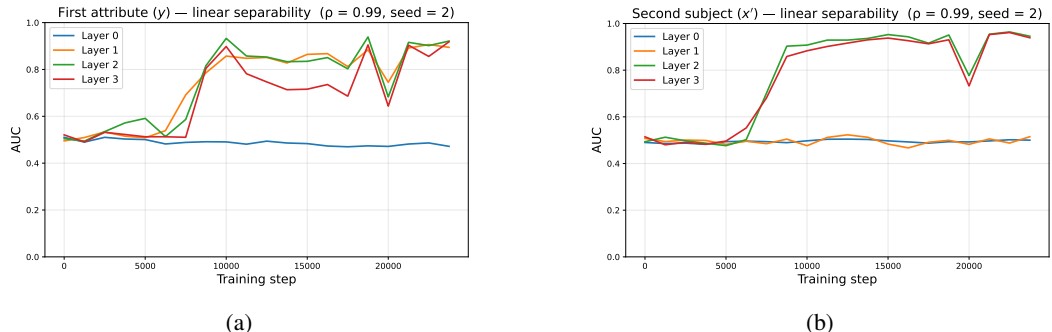

Figure 10: Linear separability across layers for a 3-layer model; linear separability on the $x'$ token is created after *copying* the signal from the $y$ token in the second layer.

copies the signal and create a linear separation that persists the last layer. This is the mechanism that emerges in 4/5 random initializations of a 3-layer model, and is clearly manifested in the attention patterns (fig. 9) and in the linear classification accuracy across layers (fig. 10).

### D.1 Bridging the gap between the fully-trainable model and the toy model.

Our theoretical analysis (appendix E) is motivated by the structured patterns that emerge in the attention kernel—the $OV$ matrix—when it is visualized (fig. 1). To test whether a comparable mechanism appears when we employ dense embeddings and allow the $KV$ matrices to train freely (thus removing the enforced uniform attention over $x, y$), we train a model with a large hidden dimension but only a small set of facts to memorize ($|\mathcal{S}| = 32$ and $d_{\text{model}} = 512$). We freeze the randomly-initialized dense embeddings and train all other parameters. The limited number of subjects makes the memorization patterns easier to inspect, while the high dimensionality approximates the regime of mutually orthogonal embeddings required by the theory.

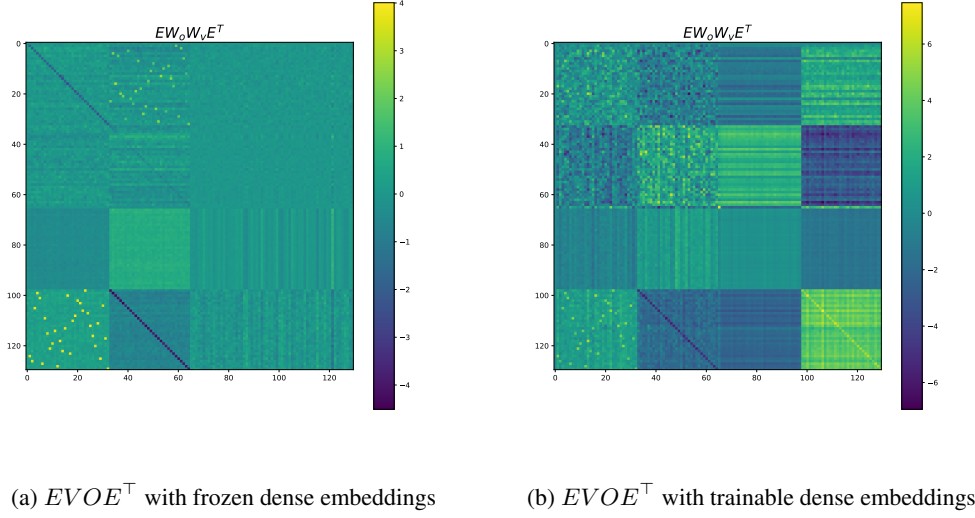

(a) $EVOE^\top$ with frozen dense embeddings      (b) $EVOE^\top$ with trainable dense embeddings

Figure 11: Visualization of the attention matrix with dense embeddings.

Because the model now uses dense embeddings—so individual coordinates no longer correspond directly to vocabulary items—we do not expect an obvious block structure in the raw $OV$ matrix. Instead, following Dar et al. [2023], we visualize $EVOE^\top$, where $E$ concatenates the input and output embedding matrices. This operation computes the pairwise similarities between embeddings as induced by the $VO$ transformation. Concretely, $(EVOE^\top)_{ij} = E_i^\top V, O, E_j$ measures how strongly the value vector elicited by symbol $i$ aligns with the output direction that scores symbol $j$, so every cell again describes a relation between concrete symbols, exactly what the raw $OV$ matrix showed when the embeddings were one-hot. The resulting heat-map (fig. 11a) exhibits a strikingly similar pattern to that observed with frozen one-hot embeddings and a fixed attention pattern, suggesting that the dense model converges to a similar underlying mechanism. In contrast, when we do train the embeddings, the pattern partially disappears, as parts of the memorization can occur in the embeddings themselves (fig. 11b). In general, there is much more variability between runs and hyperparameters when training the embeddings, where some hyperparameter choices do not show a pattern that is highly similar to the idealized one.

With a full set of $|\mathcal{S}| = d_{\text{model}} = 512$ tokens, the global pattern is hard to spot at first glance. If we instead sub-sample 28 $x$ tokens, retain only their partners $g(x)$, and then sort the rows/columns, the latent memorization re-emerges: the lower-left block collapses into a clear diagonal (the previously random pattern in the leftmost lower block in fig. 11a is transformed into a diagonal due to the sorting). This diagonal appears whether the embeddings are frozen or trainable (see figs. 12a and 12b).

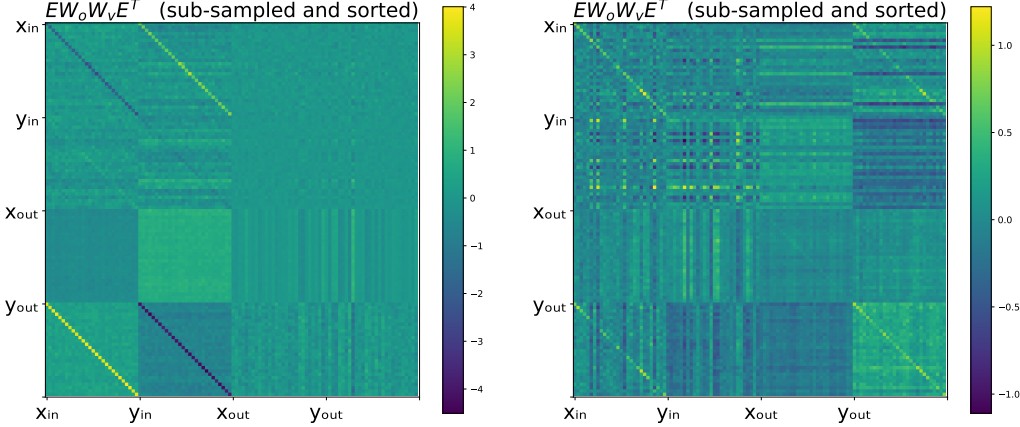

(a) $EVOE^\top$ with frozen dense embeddings (sub-sampled and sorted)

(b) $EVOE^\top$ with trainable dense embeddings (sub-sampled and sorted)

Figure 12: Visualization of the attention matrix with dense embeddings.

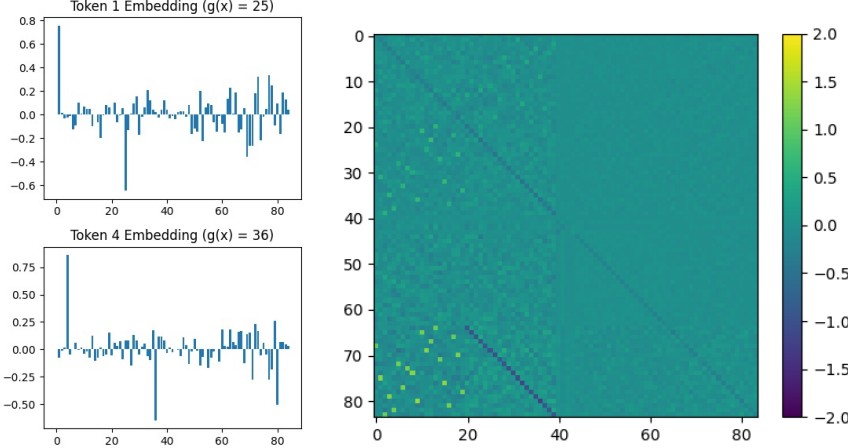

Figure 13: Visualization of learned embeddings and value matrix for a model as in Section 4 with learned embeddings, initialized to one-hot.

**One possible circuit with learned embeddings.** We now present one possible circuit that we found when initializing with the one-hot embeddings, in a simplified architecture with uniform attention as in Section 4. We still denote $e_x, e_y, u_x, u_y$ the one-hot embeddings as in Section 4, which only refer to the initialization in this setting with learned embeddings. After training, we may visualize the learned embeddings and interpret them as linear combinations of the initial one-hot embeddings, as shown in Figure 13. Denoting $\tilde{e}_x, \tilde{e}_y, \tilde{u}_x, \tilde{u}_y$ the embeddings after training, the circuit we found

looks as follows:

$$\tilde{e}_x = e_x - e_{g(x)}$$
$$\tilde{e}_y = e_y - e_{g^{-1}(y)}$$
$$\tilde{u}_x = \sum_x u_x - \sum_y u_y$$
$$\tilde{u}_y = u_y + e_{g^{-1}(y)}$$
$$W = \sum_x (u_{g(x)} - e_x)e_x^\top - \sum_y (e_y + u_y)e_y^\top.$$

The approximation $\tilde{e}_x = e_x - e_{g(x)}$, for instance, follows from the two large positive and negative spikes in the left part of fig. 13, for indices 1 and 25/36. Similar to our analysis of Section 4, we compute the quantity $W(\tilde{e}_x + \tilde{e}_y)$, which appears in the residual stream for both token $y$ and token $x'$:

$$W(\tilde{e}_x + \tilde{e}_y) = u_{g(x)} - e_x + e_{g(x)} + u_{g(x)} - e_y - u_y - u_y + e_{g^{-1}(y)}$$

We observe that this vanishes when $y = g(x)$, suggesting that a similar mechanism as in the fixed embeddings case studied in Section 4 is at play, where layer-norm can lead to sharper predictions for true sequences, as well as provide a truth direction.

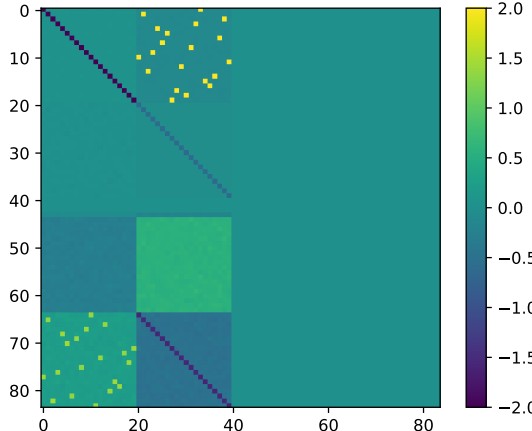

Figure 14: Structure of the value matrix $W$ when training without positional embeddings.

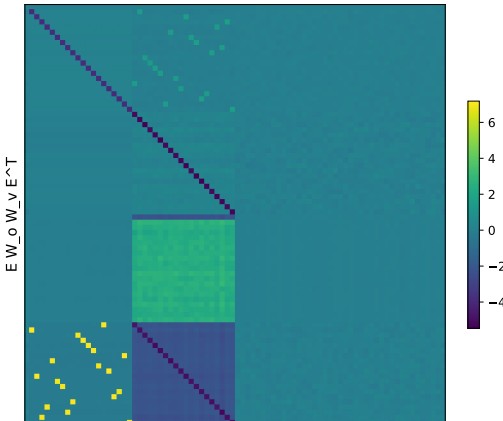

Figure 15: Structure of the value matrix $W$ when training with euclidean normalization.

# E  Theoretical analysis

This section contains theoretical analysis and proofs for the results in Section 4.

## E.1  Training dynamics

We now provide some theoretical insights on the training dynamics in the simple one-layer model of Section 4. We further simplify the model here by removing positional embeddings. Figure 14 shows that the model still learns the relevant blocks even without positional embeddings, though some of the uniform distributions on unembeddings are now absorbed in other blocks.

The lemma below highlights the structure of the gradient for a softmax classification model consisting of a linear model followed by a layer-norm operation.

**Lemma 1.** *Consider the model $F_W(x) = U \cdot \mathsf{N}(a_x + Wb_x) \in \mathbb{R}^{2N}$, with $\mathsf{N}(v) = v/\|v\|$, and the following cross-entropy population loss on some distribution over $(x, y)$:*

$$L(W) = \mathbb{E}_{x,y}[-\log \mathcal{S}(F_W(x))_y], \tag{27}$$

*where $y$ is the label and $\mathcal{S}$ the softmax operation. The gradient with respect to $W$ is then given by:*

$$\nabla L(W) = \sum_{k=1}^{2N} \mathbb{E}_{x,y}\left[\frac{\mathcal{S}(U \cdot \mathsf{N}(v_x))_k - \mathbf{1}\{y = k\}}{\|v_x\|}\mathsf{P}_{(v_x/\|v_x\|)}u_k b_x^\top\right], \tag{28}$$

with $v_x = a_x + Wb_x$ and where $\mathsf{P}_\theta = I - \theta\theta^\top$ is the projection onto the tangent space at $\theta \in \mathbb{S}^d$.

Let us decompose the population loss as

$$L(W) = L_1(W) + L_2(W) + L_3(W), \tag{29}$$

where $L_t(W)$ is the next-token prediction loss for predicting $z_{t+1}$ from $z_{1:t}$, with $z_{1:4} = (x, y, x', y')$. We show the following result.

**Theorem 3.** *Consider the following algorithm, with step-size $\eta = N/\rho$, and initialization $W_0 = 0$:*

1. *Set $W_1 = W_0 - \eta\nabla L_1(W_0)$*

2. *Set $W_2 = W_1 - \eta\nabla L_1(W_1)$*

3. *Set $W_3 = W_2 - \eta\nabla L_3(W_2)$*

*Then, we have*

$$W_3 = \sum_{x=1}^{N} \left(\beta_1 u_{g(x)} - \alpha_1 e_x\right) e_x^\top + \sum_y (\alpha_2 e_{g^{-1}(y)} - \beta_2 u_y) e_y^\top + O_\infty(1/N), \tag{30}$$

*where $\alpha_1, \alpha_2, \beta_1, \beta_2 > 0$ can be found in the proof, and $O_\infty(1/N)$ is a matrix where all entries are $O(1/N)$.*

**Comment: Euclidean norm vs. RMS Norm.** The updates in this section are derived under Euclidean layer-norm $N(v) = v/\|v\|_2$, so the normalized scores entering the softmax are attenuated (compared to our experiments which use inverse temperature $\beta = \sqrt{d} = \Theta(\sqrt{N})$) by a factor of order $1/\sqrt{N}$. Consequently, in the early regime with a fixed unembedding $U$ and $\Theta(N)$ competing classes, the correct-token probability $\sigma_{x,g(x)}$ is $O(1/N)$. In our implementation we use RMSNorm, $N_{\mathrm{RMS}}(v) = \sqrt{d}\,\frac{v}{\|v\|_2}$, which is equivalent to keeping Euclidean LN but applying a softmax with inverse temperature $\beta = \sqrt{d}$. Since $d = \Theta(N)$ (here $d = 4N + 3$), this multiplies every Euclidean score difference by $\sqrt{d} = \Theta(\sqrt{N})$, so relative advantages that were $O(1/\sqrt{N})$ become $\Theta(1)$. As a result, in the same early regime the correct-token probability becomes $\Theta(1)$. Empirically, we see similar structures emerge in the matrix during early training when using the euclidean norm instead of the RMS norm (fig. 15).

*Proof.* Let us decompose each loss into contributions from true and false sequences, which follows from the fact that the data distribution is a mixture of the two:

$$L_i(W) = \rho L_i^T(W) + (1 - \rho)L_i^F(W).$$

**Step 1.** In the first step, we take a gradient step only on the loss $L_1$ for the prediction of the second token $y$ at the first token $x$, starting from initialization $W_0 = 0$. Recall that this model takes the form $F(x) = U \cdot \mathsf{N}(e_x + We_x)$, so that in the notation of Lemma 1 we have $a_x = b_x = v_x = e_x$ and $P_{v_x/\|v_x\|}u_k = u_k$. Note that since the logits are all zero, we have $\mathcal{S}(0)_k = \frac{1}{2N}$.

We begin with the gradient on true sequences. Multiplying eq. (28) by $-\eta$ and setting $y = g(x)$ gives

$$-\eta\nabla L_1^T(W_0) = \eta\mathbb{E}_x[u_{g(x)}e_x^\top] - \eta\sum_{k=1}^{2N} \mathcal{S}(0)_k u_k \mathbb{E}_x[e_x^\top]$$

$$= \frac{\eta}{N}\sum_{x=1}^{N} u_{g(x)}e_x^\top - \frac{\eta}{2N^2}\sum_{z=1}^{2N}\sum_{x=1}^{N} u_z e_x^\top$$

$$= \frac{\eta}{N}\sum_{x=1}^{N} u_{g(x)}e_x^\top + O_\infty(\eta/N^2).$$

On false sequences, we have

$$-\eta\nabla L_1^F(W_0) = -\eta\mathbb{E}_x\Big[\sum_{k=1}^{2N}\mathcal{S}(0)_k u_k e_x^\top\Big] + \eta\mathbb{E}_{x,y}[u_y e_x^\top]$$

$$= \frac{\eta}{N^2}\sum_{x=1}^{N}\sum_{y=N+1}^{2N} u_y e_x^\top - \frac{\eta}{2N^2}\sum_{z=1}^{2N}\sum_{x=1}^{N} u_z e_x^\top$$

$$= O_\infty(\eta/N^2),$$

using the fact that $x$ and $y$ are independent. With $\eta = N/\rho$, we obtain

$$W_1 = W_0 - \eta\nabla L_1(W_0) = \sum_{x=1}^{N} u_{g(x)} e_x^\top + O_\infty(1/N).$$

**Step 2.** The second step is taken at $W = W_1 = \sum_{x=1}^{N} u_{g(x)} e_x^\top + R$, with $\|R\|_\infty = O(1/N)$. Thus, we have $v_x = e_x + W_1 e_x = e_x + u_{g(x)} + \varepsilon_x$, with $\|\varepsilon_x\|_\infty = O(1/N)$ since $e_x$ is one-hot, which implies $\|\varepsilon_x\|_2 = O(1/\sqrt{N})$. We also denote $\sigma_{x,k} := \mathcal{S}(U\cdot\mathsf{N}(v_x))_k$, which satisfies $\sigma_{x,k} = O(1/N)$ for all $x$ and $k$, since $\exp(u_k^\top\mathsf{N}(v_x)) = \Theta(1)$ for all $x$ and $k$. On true sequences, we have:

$$-\eta\nabla L_1^T(W_1) = \frac{\eta}{N}\sum_{x=1}^{N}\frac{1}{\|v_x\|_2}\Big(I - \frac{v_x v_x^\top}{\|v_x\|_2^2}\Big) u_{g(x)} e_x^\top - \frac{\eta}{N}\sum_{x=1}^{N}\sum_{k=1}^{2N}\frac{\sigma_{x,k}}{\|v_x\|_2}\Big(I - \frac{v_x v_x^\top}{\|v_x\|_2^2}\Big) u_k e_x^\top.$$

Note that we have

$$\|v_x\| = \sqrt{2 + (e_x + u_{g(x)})^\top\varepsilon_x + \|\varepsilon_x\|_2^2} = \sqrt{2 + O(1/N)} = \sqrt{2} + O(1/N).$$

by Taylor expansion. Then,

$$v_{x,k} := \frac{1}{\|v_x\|_2}\Big(I - \frac{v_x v_x^\top}{\|v_x\|_2^2}\Big) u_k = \frac{1}{\sqrt{2} + O(\frac{1}{N})} u_k + \frac{\delta_{k,g(x)} + O(\frac{1}{N})}{2\sqrt{2} + O(\frac{1}{N})} v_x$$

$$= \frac{1}{\sqrt{2}} u_k + \frac{\delta_{k,g(x)}}{2}(e_x + u_{g(x)}) + O_\infty(1/N),$$

where $O_\infty(1/N)$ is a vector with $\ell_\infty$ norm $O(1/N)$ and $\delta_{k,g(x)} = \mathbf{1}\{k = g(x)\}$ denotes the Kronecker delta. Plugging back into the gradient above, we obtain

$$-\eta\nabla L_1^T(W_1) = \frac{\eta}{N\sqrt{2}}\sum_{x=1}^{N}\Big(u_{g(x)} - \frac{1}{2}(e_x + u_{g(x)}) + O_\infty\Big(\frac{1}{N}\Big)\Big) e_x^\top$$

$$- \frac{\eta}{N\sqrt{2}}\sum_{x=1}^{N}\sum_{k=1}^{2N}\sigma_{x,k} v_{x,k} e_x^\top$$

$$= \frac{\eta}{2\sqrt{2}N}\sum_{x=1}^{N}(u_{g(x)} - e_x) e_x^\top + O_\infty(\eta/N^2),$$

which follows by noticing that $\sum_{k=1}^{2N}\sigma_{x,k} v_{x,k} = O_\infty(1/N)$. For false sequences, we have

$$-\eta\nabla L_1^F(W_1) = \eta\mathbb{E}_{x,y}[v_{x,y} e_x^\top] - \frac{\eta}{N}\sum_{x=1}^{N}\sum_{k=1}^{2N}\sigma_{x,k} v_{x,k} e_x^\top$$

$$= \frac{\eta}{N^2}\sum_{x=1}^{N}\sum_{y=N+1}^{2N} v_{x,y} e_x^\top - \frac{\eta}{N}\sum_{x=1}^{N}\sum_{k=1}^{2N}\sigma_{x,k} v_{x,k} e_x^\top$$

$$= O_\infty(\eta/N^2).$$

This again follows by noticing that

$$\frac{1}{N}\sum_{y=N+1}^{2N} v_{x,y} = O_\infty(1/N) \quad\text{and}\quad \sum_{k=1}^{2N} v_{x,k} = O_\infty(1/N).$$

With $\eta = N/\rho$, this yields

$$W_2 = W_1 - \eta \nabla L_1(W_1) = \sum_{t=1}^{N} \left( \alpha u_{g(t)} - e_t \right) e_t^\top + O(1/N),$$

with $\alpha = 1 + \frac{1}{2\sqrt{2}}$.

**Step 3.** The third step takes one gradient step on the loss $L_3$ at the third token, i.e., predicting $y'$ from $(x, y, x')$. The model now takes the form $F(x, y, x') = U \cdot \mathsf{N}(e_{x'} + \frac{1}{3}W(e_x + e_y + e_{x'}))$, with a uniform attention on the first three tokens.

We get the gradient of the loss on $y'$ from (28), giving[5]

$$v_{x,y,x'} = e_{x'} + \frac{1}{3}W_2(e_x + e_y + e_{x'})$$

$$= \frac{2}{3}e_{x'} - \frac{1}{3}e_x + \frac{\alpha}{3}u_{g(x)} + \frac{\alpha}{3}u_{g(x')} + \varepsilon_{x,y,x'} =: v_{x,x'} + \varepsilon_{x,y,x'},$$

with $\|\varepsilon_{x,y,x'}\|_\infty = O(1/N)$, since it is the sum of 3 columns of the $O_\infty(1/N)$ term in $W_2$. As in the second step, we have $\|\varepsilon_{x,y,x'}\|_2^2 = O(1/N)$ and $v_{x,x'}^\top \varepsilon_{x,y,x'} = O(1/N)$, so that by Taylor expansion we have $\|v_{x,y,x'}\| = \|v_{x,x'}\| + O(1/N) = \frac{1}{3}\sqrt{5 + 2\alpha^2} + O(1/N)$ for $x \neq x'$ and $\|v_{x,y,x'}\| = \|v_{x,x'}\| + O(1/N) = \frac{1}{3}\sqrt{1 + 2\alpha^2} + O(1/N)$ for $x = x'$. Note that we once again have $\sigma_{x,y,x',k} := \mathcal{S}(U \cdot \mathsf{N}(v_{x,y,x'}))_k = \tilde{O}(1/N)$ due to the normalization. On true sequences, we have

$$-\eta \nabla L_3^T(W_2) = \eta \mathbb{E}_{x,x'}\left[ \frac{1}{3\|v_{x,y,x'}\|} \left( I - \frac{v_{x,y,x'}v_{x,y,x'}^\top}{\|v_{x,y,x'}\|^2} \right) u_{g(x')}(e_x + e_{g(x)} + e_{x'})^\top \right]$$

$$- \eta \sum_{k=1}^{2N} \mathbb{E}_{x,x'}\left[ \frac{\sigma_{x,g(x),x',k}}{3\|v_{x,y,x'}\|} \left( I - \frac{v_{x,y,x'}v_{x,y,x'}^\top}{\|v_{x,y,x'}\|^2} \right) u_k(e_x + e_{g(x)} + e_{x'})^\top \right].$$

Let us first show that the error terms $\varepsilon_{x,y,x'}$ lead to negligible contributions to the gradient. Note that we have

$$v_{x,y,x',k} := \frac{1}{3\|v_{x,y,x'}\|} \left( I - \frac{v_{x,y,x'}v_{x,y,x'}^\top}{\|v_{x,y,x'}\|^2} \right) u_k$$

$$= \frac{1}{3(\|v_{x,x'}\| + O(\frac{1}{N}))} \left( I - \frac{(v_{x,x'} + O_\infty(\frac{1}{N}))(v_{x,x'}^\top + O_\infty(\frac{1}{N}))}{\|v_{x,x'}\|^2 + O(\frac{1}{N})} \right) u_k$$

$$= \frac{1}{3\|v_{x,x'}\|} \left( I - \frac{v_{x,x'}v_{x,x'}^\top}{\|v_{x,x'}\|^2} \right) u_k + \varepsilon_{x,y,x',k},$$

With $\|\varepsilon_{x,y,x',k}\|_\infty = O(1/N)$, where we used $\frac{1}{a+\epsilon} = \frac{1}{a} + O(\epsilon)$ for $a > 0$, and $(u + \epsilon)(u^T + \epsilon) = uu^T + O_\infty(1/N)$ for $\epsilon = O_\infty(1/N)$.

Then, taking expectations with respect to independent $x, x'$, it is easy to check that

$$\mathbb{E}_{x,x'}[\varepsilon_{x,g(x),x',g(x')}(e_x + e_{g(x)} + e_{x'})^\top] = O_\infty(1/N^2)$$

$$\mathbb{E}_{x,x'}[\sigma_{x,g(x),x',k}\varepsilon_{x,g(x),x',k}(e_x + e_{g(x)} + e_{x'})^\top] = O_\infty(1/N^3).$$

The gradient update can then be rewritten as

$$-\eta \nabla L_3^T(W_2) = \eta \mathbb{E}_{x,x'}\left[ \frac{1}{3\|v_{x,x'}\|} \left( I - \frac{v_{x,x'}v_{x,x'}^\top}{\|v_{x,x'}\|^2} \right) u_{g(x')}(e_x + e_{g(x)} + e_{x'})^\top \right] \tag{31}$$

$$- \eta \sum_{k=1}^{2N} \mathbb{E}_{x,x'}\left[ \frac{\sigma_{x,g(x),x',k}}{3\|v_{x,x'}\|} \left( I - \frac{v_{x,x'}v_{x,x'}^\top}{\|v_{x,x'}\|^2} \right) u_k(e_x + e_{g(x)} + e_{x'})^\top \right] \tag{32}$$

$$+ O_\infty(\eta/N^2) \tag{33}$$

---

[5] We use the fact $W_2 e_t = \alpha u_{g(t)} - e_t + O_\infty(1/N)$ for $t \leq N$ and $O_\infty(1/N)$ otherwise.

We now check that the second term is also of order $O_\infty(\eta/N^2)$. Indeed, the projector matrix $\frac{1}{3\|v_{x,x'}\|}\left(I - \frac{v_{x,x'}v_{x,x'}^\top}{\|v_{x,x'}\|^2}\right)$ is sparse with rows of bounded $\ell_1$ norm, and we have $\sum_k \sigma_{x,g(x),x',k} u_k = O_\infty(1/N)$, so that

$$\sum_{k=1}^{2N} \frac{\sigma_{x,g(x),x',k}}{3\|v_{x,x'}\|}\left(I - \frac{v_{x,x'}v_{x,x'}^\top}{\|v_{x,x'}\|^2}\right) u_k =: \zeta_{x,x'} = O_\infty(1/N),$$

for all $x, x'$. We then have

$$\sum_{k=1}^{2N} \mathbb{E}_{x,x'}\left[\frac{\sigma_{x,g(x),x',k}}{3\|v_{x,x'}\|}\left(I - \frac{v_{x,x'}v_{x,x'}^\top}{\|v_{x,x'}\|^2}\right) u_k(e_x + e_{g(x)} + e_{x'})^\top\right]$$

$$= \mathbb{E}_{x,x'}[\zeta_{x,x'}(e_x + e_{g(x)} + e_{x'})^\top]$$

$$= \frac{1}{N}\sum_{x=1}^N \zeta_{x,x'} e_x^\top + \frac{1}{N}\sum_{x=1}^N \zeta_{x,x'} e_{g(x)}^\top + \frac{1}{N}\sum_{x=1}^N \zeta_{x,x'} e_{x'}^\top = O_\infty(1/N^2).$$

For the first term (31), we have

$$\eta\mathbb{E}_{x,x'}\left[\frac{1}{3\|v_{x,x'}\|}\left(I - \frac{v_{x,x'}v_{x,x'}^\top}{\|v_{x,x'}\|^2}\right) u_{g(x')}(e_x + e_{g(x)} + e_{x'})^\top\right]$$

$$\overset{*}{=} \eta\mathbb{E}_{x,x'}\left[\frac{1}{3\|v_{x,x'}\|} u_{g(x')} e_{x'}^\top\right] + O_\infty(\eta/N^2) - \eta\mathbb{E}_{x,x'}\left[\frac{\alpha(1 + \delta_{g(x),g(x')})}{9\|v_{x,x'}\|^3} v_{x,x'}(e_x + e_{g(x)} + e_{x'})^\top\right]$$

$$= \frac{\eta\beta_1}{N}\sum_{x=1}^N u_{g(x)} e_x^\top - \eta\mathbb{E}_{x,x'}\left[\gamma_{x,x'} v_{x,x'}(e_x + e_{g(x)} + e_{x'})^\top\right] + O_\infty(\eta/N^2),$$

with

$$\beta_1 = \mathbb{E}_x\left[\frac{1}{3\|v_{x,1}\|}\right] \quad \text{and} \quad \gamma_{x,x'} = \frac{\alpha(1 + \delta_{g(x),g(x')})}{9\|v_{x,x'}\|^3}.$$

In $(*)$ we used (i) that $\mathbb{E}_{x,x'}\left[\frac{1}{3\|v_{x,x'}\|} u_{g(x')}(e_x + e_{g(x)})^\top\right] = O_\infty(1/N^2)$ thanks to the independence of $x$ and $x'$; and (ii) the fact that $u_{g(x')}^T v_{x,x'} = \frac{\alpha}{3}(1 + \delta_{g(x),g(x')})$ by definition of $v_{x,x'}$ and thanks to orthogonality.

We have

$$-\eta\mathbb{E}_{x,x'}\left[\gamma_{x,x'} v_{x,x'}(e_x + e_{g(x)} + e_{x'})^\top\right]$$

$$= -\eta\mathbb{E}_x[\mathbb{E}_{x'}[\gamma_{x,x'} v_{x,x'}|x](e_x + e_{g(x)})^\top] - \eta\mathbb{E}_{x'}[\mathbb{E}_x[\gamma_{x,x'} v_{x,x'}|x']e_{x'}^\top]$$

$$\overset{*}{=} \frac{\eta\beta_2}{N}\sum_{x=1}^N (e_x - \alpha u_{g(x)})(e_x + e_{g(x)})^\top - \frac{\eta\beta_2}{N}\sum_{x=1}^N (2e_x + \alpha u_{g(x)})e_x^\top + O(\eta/N^2)$$

$$\overset{**}{=} -\frac{\eta\beta_2}{N}\sum_{x=1}^N e_x e_x^\top + \frac{\eta\beta_2}{N}\sum_{y=N+1}^{2N} (e_{g^{-1}(y)} - \alpha u_y)e_y^\top + O_\infty(\eta/N^2),$$

with

$$\beta_2 = \frac{1}{3}\mathbb{E}_{x'}[\gamma_{1,x'}] = \frac{1}{3}\mathbb{E}_x[\gamma_{x,1}] = \frac{1}{3N}\gamma_{1,1} + \frac{N-1}{3N}\gamma_{1,2}.$$

In $(*)$ we condition on $x$ and decompose $v_{x,x'} = A(x) + B(x')$, where $A(x)$ is independent of $x'$; then linearity gives $\mathbb{E}_{x'}[\gamma_{x,x'} v_{x,x'} \mid x] = A(x)\,\mathbb{E}_{x'}[\gamma_{x,x'}] + \mathbb{E}_{x'}[\gamma_{x,x'} B(x')]$. By permutation symmetry of labels, $\mathbb{E}_{x'}[\gamma_{x,x'}] = \mathbb{E}_z[\gamma_{1,z}] = 3\beta_2$, while the $B(x')$ part averages over a uniformly random index and is $O_\infty(1/N)$; the same holds with $x$ and $x'$ swapped for the second bracket. Substituting these two conditionals into the split expectation, replacing outer expectations by uniform

sums, and renaming the dummy index yields the displayed $(*)$ line. In $(**)$ we perform change of variables $y = g(x)$. We have thus shown

$$-\eta \nabla L_3^T(W_2) = \frac{\eta}{N} \sum_{x=1}^{N} (\beta_1 u_{g(x)} - \beta_2 e_x) e_x^\top + \frac{\eta \beta_2}{N} \sum_{y=N+1}^{2N} (e_{g^{-1}(y)} - \alpha u_y) e_y^\top + O_\infty(\eta/N^2). \quad (34)$$

For false sequences, it can be checked that $\eta \nabla L_3^F(W_2) = O_\infty(\eta/N^2)$. Thus, taking step-size $\eta = N/\rho$ yields

$$W_3 = W_2 - \eta \nabla L_3(W_2)$$

$$= (\alpha + \beta_1) \sum_{x=1}^{N} u_{g(x)} e_x^\top - (1 + \beta_2) \sum_{x=1}^{N} e_x e_x^\top + \beta_2 \sum_{y=N+1}^{2N} (e_{g^{-1}(y)} - \alpha u_y) e_y^\top + O_\infty(1/N).$$

$\square$

### E.1.1 Learning (positional) attention.

We now turn to learning the key–query matrix under positional attention, assuming that the value matrix has already been learned with the structure described above. Specifically, we show that the gradient of the key–query matrix on true sequences drives positional attention to focus on the $x'$ token, effectively causing the model to ignore the initial $(x, y)$ pair. This observation may account for the absence of emergence at $\rho = 1$, although it does not explain why emergence still occurs when attention is trainable and $\rho < 1$.

For this part, assume a simple architecture of the following form for the prediction of the fourth token logits given the first three tokens:

$$F_{W_{KQ}}(z_{1:3}) = U \cdot \sum_{t=1}^{3} \frac{\exp(p_t^\top W_{KQ} p_3)}{\sum_{t'=1}^{3} \exp(p_{t'}^\top W_{KQ} p_3)} W_V e_{z_t},$$

where $z_{1:3} = (e_x, e_y, e_{x'})$, and $p_{1:3}$ are the positional embeddings defined in (2).

We assume $W_V$ fixed to the the structure in (6)-(7), with $\beta_1 = \beta_2 =: \beta$ for simplicity. We consider the following population loss for $W_{KQ}$ on the last token for true sequences:

$$L(W_{KQ}) = \mathbb{E}_{x,x'}[\ell(g(x'), F_{W_{KQ}}(x, g(x), x'))].$$

Then, the negative gradient direction at $W_{KQ} = 0$ is given by

$$-\nabla L(W_{KQ}) = \frac{1}{3} \sum_{t=1}^{3} \sum_{k=1}^{2N} \mathbb{E}_{x,x'}[(\mathbf{1}\{g(x') = k\} - \hat{p}(k|x, x')) u_k^\top W_V e_{z_t} (p_t - \bar{p}_{1:3}) p_3^\top], \quad (35)$$

where we denote $z_{1:3} = (x, g(x), x')$ and $\bar{p}_{1:3} = \frac{1}{3}(p_1 + p_2 + p_3)$, and $\hat{p}$ are probability predictions at $W_{KQ} = 0$, which we assume satisfy the following, given the assumed structure on $W_V$, for some small $\epsilon$:[6]

$$\hat{p}(k|x, x') = \begin{cases} (1 - \epsilon)/2, & \text{if } k \in \{g(x), g(x')\} \text{ and } x \neq x', \\ 1 - \epsilon, & \text{if } k = g(x) \text{ and } x = x', \\ O(1/N), & \text{o/w.} \end{cases}$$

Let us now write the update in (35) as

$$-\nabla L(W_{KQ}) = \frac{1}{3} \sum_{t=1}^{3} w_t (p_t - \bar{p}_{1:3}) p_3^\top,$$

---

[6]This requires that we the early phase is run for long enough so that $\beta$ is large enough, say $O(\log N)$.

and study the values of the $w_t$. For $t = 1$, we have

$$w_1 = \sum_{k=1}^{2N} \mathbb{E}_{x,x'}[(\mathbf{1}\{g(x') = k\} - \hat{p}(k|x,x'))u_k^\top W_V e_x]$$

$$= \beta \sum_{k=1}^{2N} \mathbb{E}_{x,x'}[(\mathbf{1}\{g(x') = k\} - \hat{p}(k|x,x'))\mathbf{1}\{g(x) = k\}]$$

$$= \beta \sum_{k=N+1}^{2N} \mathbb{E}_x[\mathbf{1}\{g(x) = k\}]\mathbb{E}_{x'}[\mathbf{1}\{g(x') = k\}] - \beta\mathbb{E}_{x,x'}[\hat{p}(g(x)|x,x')]$$

$$= \frac{\beta}{N} - \frac{\beta}{N}(1 - \epsilon) - \frac{\beta(N^2 - N)}{N^2}\frac{1 - \epsilon}{2} \leq -\beta\frac{1 - \epsilon}{2} + O(\beta/N).$$

For $t = 2$, we have

$$w_2 = \sum_{k=1}^{2N} \mathbb{E}_{x,x'}[(\mathbf{1}\{g(x') = k\} - \hat{p}(k|x,x'))u_k^\top W_V e_{g(x)}]$$

$$= -\beta \sum_{k=1}^{2N} \mathbb{E}_{x,x'}[(\mathbf{1}\{g(x') = k\} - \hat{p}(k|x,x'))\mathbf{1}\{x = k\}]$$

$$= 0 + \beta\mathbb{E}_{x,x'}[\hat{p}(x|x,x')]$$

$$= O(\beta/N).$$

For $t = 3$, we have

$$w_3 = \sum_{k=1}^{2N} \mathbb{E}_{x,x'}[(\mathbf{1}\{g(x') = k\} - \hat{p}(k|x,x'))u_k^\top W_V e_{x'}]$$

$$= \beta \sum_{k=1}^{2N} \mathbb{E}_{x,x'}[(\mathbf{1}\{g(x') = k\} - \hat{p}(k|x,x'))\mathbf{1}\{g(x') = k\}]$$

$$= \beta - \mathbb{E}_{x,x'}[\hat{p}(g(x')|x,x')]$$

$$\geq \beta - \beta\frac{1 - \epsilon}{2} \geq \frac{\beta}{2}$$

Thus, when $N \gg \beta$, we have $w_3 - w_1, w_3 - w_2 \geq \beta/2 + O(\beta/N)$. Taking $W_{KQ}^1 = -\eta\nabla L$, the gap in attention logits between $t = 3$ and $t = 1, 2$ is of order $\eta\beta/6$, so that for $\eta$ large enough, the attention mostly focuses on the third token $x'$.

## E.2 Proof of Theorem 1

Suppose we are given $(x, y, x')$, where we assume for simplicity that $x \neq x'$ and $g(x') \neq y$. Denote by $f_W(z_{1:t})$ the output of the model in (4) before applying the LN and the unembedding layer. Then, we have that:

$$f_W(x, y, x') = e_{x'} + p_3 + \frac{1}{3}\bar{\gamma}\left(\sum_y u_y - \sum_x u_x\right) +$$

$$+ \frac{1}{3}\left(-\alpha_1 e_x + \beta_1 u_{g(x)} + \alpha_2 e_{g^{-1}(y)} - \beta_2 u_y - \alpha_1 e_{x'} + \beta_1 u_{g(x')}\right) \quad (36)$$

Denote by $c_1 := 2 + \frac{\bar{\gamma}^2(2N-2)+2\alpha_1^2+\beta_1^2}{9}$ and $c_2 := 2 + \frac{\bar{\gamma}^2(2N-3)+2\alpha_1^2+\beta_1^2}{9}$ . for a true sample where $y = g(x)$ we have that:

$$\|f_W(x, g(x), x')\|^2 = c + (\beta_1 - \beta_2 + \bar{\gamma})^2 + (\beta_1 + \bar{\gamma})^2 .$$

Hence, after applying the LN and unembedding layer we have that:

$$(F_W(x, g(x), x'))_{g(x')} = \frac{\beta_1 + \bar{\gamma}}{3\sqrt{c_1 + (\beta_1 - \beta_2 + \bar{\gamma})^2 + (\beta_1 + \bar{\gamma})^2}}$$

$$\max_{y' \neq g(x')} (F_W(x, g(x), x'))_{y'} = \frac{\bar{\gamma} + \max(0, \beta_1 - \beta_2)}{3\sqrt{c_1 + (\beta_1 - \beta_2 + \bar{\gamma})^2 + (\beta_1 + \bar{\gamma})^2}}$$

For a false sample where $y \neq g(x)$ we have that:

$$\|f_W(x, g(x), x')\|^2 = c_2 + 2(\beta_1 + \bar{\gamma})^2 + (-\beta_2 + \bar{\gamma})^2 .$$

Hence, after applying the LN and unembedding layer we have that:

$$(F_W(x, y, x'))_{g(x')} = \frac{\beta_1 + \bar{\gamma}}{3\sqrt{c_2 + 2(\beta_1 + \bar{\gamma})^2 + (-\beta_2 + \bar{\gamma})^2}}$$

$$\max_{y' \neq g(x')} (F_W(x, y, x'))_{y'} = \frac{\beta_1 + \bar{\gamma}}{3\sqrt{c_2 + 2(\beta_1 + \bar{\gamma})^2 + (-\beta_2 + \bar{\gamma})^2}} .$$

Plugging in these terms finishes the proof.

### E.3 Proof of Theorem 2

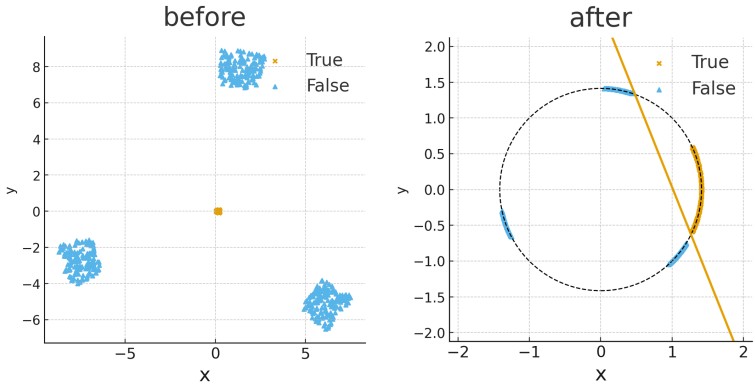

Figure 16: Illustration for a LN-induced linear separability.

*Proof.* We first describe the output of the model in (4) before applying LN. Denote by $v_T$, $v_F \in \mathbb{R}^{4N+3}$ these outputs for true and false samples respectively. Recall that a true sample $(x, y)$ is when $y = g(x)$ and false otherwise. Then, we have that:

$$v_T = e_y + p_2 + \frac{1}{2}\left((\alpha_2 - \alpha_1)e_x + (\beta_1 - \beta_2)u_y + (\gamma_1 - \gamma_2) \cdot \left(\sum_y u_y - \sum_x u_x\right)\right) \quad (37)$$

$$v_F = e_y + p_2 + \frac{1}{2}\left(-\alpha_1 e_x + \alpha_2 e_{g^{-1}(y)} + \beta_1 u_{g(x)} - \beta_2 u_y + (\gamma_1 - \gamma_2) \cdot \left(\sum_y u_y - \sum_x u_x\right)\right) \quad (38)$$

We will first show that without adding N the samples above cannot be separated for general $x$ and $y$.

Assume otherwise, that there exists a linear separator $w = \begin{pmatrix} w_1 \\ w_2 \\ w_3 \\ w_4 \\ w_5 \end{pmatrix}$ with $w_1, \ldots, w_4 \in \mathbb{R}^N, w_5 \in \mathbb{R}^3$

and bias term $b \in \mathbb{R}$ such that $\langle w, v_T \rangle - b \geq 0$ and $\langle w, v_F \rangle - b < 0$ for every true or false sample

respectively. We slightly abuse notation and write $\langle w_1, e_x \rangle$ as $\left\langle \begin{pmatrix} w_1 \\ 0_{3N+3} \end{pmatrix}, e_x \right\rangle$, and similarly when multiplying $w_2$ by $e_y$, $w_3$ by $u_x$, $w_4$ by $u_y$ and $w_5$ by $p_t$.

$$c := \frac{1}{2} \left\langle (\gamma_1 - \gamma_2) \cdot \left( \sum_y u_y - \sum_x u_x \right), w_3 + w_4 \right\rangle + \langle w_5, p_2 \rangle$$

the terms in the inner products that are independent of the sample. Then, using the linear separator on these four samples we have:

$$b \le (\alpha_2 - \alpha_1) \langle e_{x_i}, w_1 \rangle + \langle e_{y_i} w_2 \rangle + (\beta_1 - \beta_2) \langle u_{y_i}, w_4 \rangle + c \tag{39}$$

$$b \le (\alpha_2 - \alpha_1) \langle e_{x_j}, w_1 \rangle + \langle e_{y_j} w_2 \rangle + (\beta_1 - \beta_2) \langle u_{y_j}, w_4 \rangle + c \tag{40}$$

$$b \ge \alpha_2 \langle e_{x_i}, w_1 \rangle - \alpha_1 \langle e_{x_j}, w_1 \rangle + \langle e_{y_i}, w_2 \rangle + \beta_1 \langle u_{y_j}, w_4 \rangle - \beta_2 \langle u_{y_i}, w_4 \rangle + c \tag{41}$$

$$b \ge \alpha_2 \langle e_{x_j}, w_1 \rangle - \alpha_1 \langle e_{x_i}, w_1 \rangle + \langle e_{y_j}, w_2 \rangle + \beta_1 \langle u_{y_i}, w_4 \rangle - \beta_2 \langle u_{y_j}, w_4 \rangle + c. \tag{42}$$

Adding up (41) and (42) we have that:

$$2b - 2c \ge (\alpha_2 - \alpha_1) \langle e_{x_j}, w_1 \rangle + \langle e_{y_j} w_2 \rangle + (\beta_1 - \beta_2) \langle u_{y_j}, w_4 \rangle + \tag{43}$$

$$+ (\alpha_2 - \alpha_1) \langle e_{x_i}, w_1 \rangle + \langle e_{y_i} w_2 \rangle + (\beta_1 - \beta_2) \langle u_{y_i}, w_4 \rangle, \tag{44}$$

which is a contradiction to (39) and (40). This means that there is no linear separator, regardless of the values of the parameters, which proves the first item.

Assume there is layer normalization after the prediction as in (4). This means that the output of the model is $\frac{v}{\|v\|}$. Consider the linear predictor $w = p_2$, and a bias term $b$ that will be determined later. Then, the output of the linear predictor is exactly $\langle w, v \rangle = \frac{1}{\|v\|}$.

We will now calculate the norm of both true and false samples. For a true sample $(x, g(x))$ we have that:

$$\|v_T\|^2 = 2 + (\alpha_2 - \alpha_1)^2 + (\gamma_1 - \gamma_2)^2 \cdot (2N - 1) + (\gamma_1 - \gamma_2 + \beta_1 - \beta_2)^2. \tag{45}$$

For a negative sample $(x, y)$ with $g(x) \ne y$ we have:

$$\|v_F\|^2 = 2 + \alpha_1^2 + \alpha_2^2 + (\gamma_1 - \gamma_2)^2 \cdot (2N - 2) + (\gamma_1 - \gamma_2 + \beta_1)^2 + (\gamma_1 - \gamma_2 - \beta_2)^2. \tag{46}$$

There exists a linear separator as long as $\frac{1}{\|v_F\|} - \frac{1}{\|v_T\|} \ne 0$. Since the vectors $v_T$ and $v_F$ are both non-zero, this is equivalent to $\|v_T\|^2 \ne \|v_F\|^2$. By the above calculation, we have that:

$$\|v_F\|^2 - \|v_T\|^2$$
$$= \alpha_1^2 + \alpha_2^2 - (\alpha_1 - \alpha_2)^2 - (\gamma_1 - \gamma_2)^2 + (\gamma_1 - \gamma_2 + \beta_1)^2 + (\gamma_1 - \gamma_2 - \beta_2)^2 - (\gamma_1 - \gamma_2 + \beta_1 - \beta_2)^2$$
$$= 2\alpha_1\alpha_2 + 2\beta_1\beta_2.$$

This shows that if $2\alpha_1\alpha_2 + 2\beta_1\beta_2 \ne 0$ then we have a linear separation between true and false samples.

Further assuming that $\alpha_1 = \alpha_2$, $\beta_1 = \beta_2$, $\gamma_1 = \gamma_2$ we have that $\|v_T\|^2 = 2$ and $\|v_F\|^2 = 2 + 2\alpha_2 + 2\beta_2$. To find the optimal margin for this predictor we pick:

$$b = \frac{1}{2} \cdot \left( \frac{1}{\|v_T\|} - \frac{1}{\|v_F\|} \right) = \frac{1}{2\sqrt{2}} \left( 1 - \frac{1}{\sqrt{1 + \alpha^2 + \beta^2}} \right).$$

We will now prove that there is linear separation after predicting the $x'$ token. Using the output of the model as in (4) we get:

$$v_T = C + \frac{1}{3} \left( (\alpha_2 - \alpha_1)e_x + (\beta_1 - \beta_2)u_y - \alpha_1 e_{x'} + \beta_1 u_{g(x')} \right) \tag{47}$$

$$v_F = C + \frac{1}{3} \left( -\alpha_1 e_x + \alpha_2 u_{g^{-1}(y)} + \beta_1 u_{g(x)} - \beta_2 u_y + -\alpha_1 e_{x'} + \beta_1 u_{g(x')} \right), \tag{48}$$

where $C = e_{x'} + p_3 + \frac{\hat{\gamma}}{3} \cdot \left( \sum_y u_y - \sum_x u_x \right)$. We can now calculate:

$$\|v_T\|^2 = 2 + \frac{1}{9} \left( (\alpha_2 - \alpha_1)^2 + (\beta_1 - \beta_2 + \bar{\gamma})^2 + \alpha_1^2 + (\beta_1 + \bar{\gamma})^2 + (2N - 2)\bar{\gamma}^2 \right) \tag{49}$$

$$\|v_F\|^2 = 2 + \frac{1}{9} \left( 2\alpha_1^2 + \alpha_2^2 + 2(\beta_1 + \bar{\gamma})^2 + (\bar{\gamma} - \beta_2)^2 + (2N - 3)\bar{\gamma}^2 \right) . \tag{50}$$

We now have that:

$$\|v_F\|^2 - \|v_T\|^2 = \frac{1}{9} \cdot \left( \alpha_1^2 + \alpha_2^2 + (\beta_1 + \bar{\gamma})^2 + (\bar{\gamma} - \beta_2)^2 - (\alpha_2 - \alpha_1)^2 - (\beta_1 - \beta_2 + \bar{\gamma})^2 - \bar{\gamma}^2 \right)$$

$$= \frac{2}{9} (\alpha_1 \alpha_2 + \beta_1 \beta_2) .$$

By a similar argument to the previous case, if $\alpha_1 \alpha_2 + \beta_1 \beta_2 \neq 0$ then there is linear separation between true and false samples. Further assuming that $\alpha_1 = \alpha_2$, $\beta_1 = \beta_2$ and $\bar{\gamma} = 0$, to find the optimal margin for the predictor we pick:

$$b = \frac{1}{2} \cdot \left( \frac{1}{\|v_T\|} - \frac{1}{\|v_F\|} \right) = \frac{\alpha^2 + \beta^2}{9\sqrt{4 + \frac{8}{9}(\alpha^2 + \beta^2) + \frac{1}{27}(\alpha^2 + \beta^2)^2}} .$$

$\square$

### E.4 Evaluating checkpoints of a "real" LM

To test whether the two-phase dynamics also appear in a large model trained on open-web data, we analyzed the **Pythia-6.9B** training checkpoints released by EleutherAI. Using the COUNTERFACT dataset we construct each input by concatenating $K = 4$ factual statements whose preceding context is either entirely true or entirely false, mirroring our previous setup. For every checkpoint we measure three signals on the final statement:

- **Memorization**: percentage of cases where greedy decoding succeeds in completing the correct token.
- **Uncertainty**: entropy of the model's full-vocabulary distribution for predicting the last token; we record the difference between true and false context.
- **Linear separability**: accuracy of a linear probe trained to classify the truth value of the surrounding context.

Table 1: **Pythia-6.9B** metrics across training steps.

| step | $\Delta H$ | memorization | probe AUC |
|---|---|---|---|
| 0 | 0.001 | 0.000 | 0.383 |
| 512 | 0.006 | 0.000 | 0.435 |
| 1000 | 0.005 | 0.006 | 0.467 |
| 3000 | 0.219 | 0.242 | 0.587 |
| 5000 | 0.217 | 0.435 | 0.648 |
| 10000 | 0.286 | 0.547 | 0.667 |
| 20000 | 0.355 | 0.655 | 0.754 |
| 40000 | 0.329 | 0.727 | 0.759 |
| 60000 | 0.421 | 0.772 | 0.802 |
| 80000 | 0.419 | 0.822 | 0.799 |
| 100000 | 0.479 | 0.835 | 0.818 |
| 110000 | 0.485 | 0.849 | 0.835 |
| 120000 | 0.536 | 0.842 | 0.835 |
| 130000 | 0.565 | 0.858 | 0.783 |
| 143000 | 0.518 | 0.875 | 0.831 |

*Notes.* $\Delta H$ denotes the entropy gap between matched prompt pairs presented with false versus true context. The memorization rate is the share of instances in which the model's output distribution places the correct continuation token at the top-1 position. "Probe AUC" is the ROC-AUC of a linear classifier trained to predict the surrounding context's truth value from model activations.

**Findings.** *Early training ($\leq 1k$ steps).* $\Delta H \approx 0$; the model memorizes indiscriminately.
*Mid training (3k–80k).* Memorization jumps, then plateaus, while $\Delta H$ and probe accuracy climb steadily.
*Late training ($\geq 80k$).* Entropy separation continues to widen even after memorization saturates, mirroring Phase 2 but over a longer horizon.

Overall, this echoes the two-phase pattern observed in simpler experiments: an initial jump in memorization followed by a slower, steadier increase in entropy separation. Differences remain: the second phase is more gradual, and classification and entropy increase even *before* memorization stabilizes. We hypothesize this stems from continual exposure to new facts during training, unlike our idealized setup where all facts are seen in a single gradient step. The modest terminal memorization and classification scores are consistent with reports that the Pythia series is under-trained relative to its capacity.

Finally, in Pythia-6.9B we *do not* find evidence that layer normalization itself induces linear separability; rather, a linearly decodable truth signal emerges gradually with depth across many layers. Our aim was to advance one plausible mechanism for the phenomenon observed in pretrained LMs, not to claim uniqueness. Given the model's deeper architecture—with numerous layers and MLP blocks—and the richness of natural-language data, additional or distinct mechanisms are likely at play. A systematic study of these mechanisms is an important direction for future work.

