# OpenReview forum: "Emergence of Linear Truth Encodings in Language Models"
_NeurIPS.cc/2025/Conference — NeurIPS 2025 poster_

### Official Review · Reviewer_za5X · 2025-06-29

**Clarity:** 2
**Significance:** 2
**Originality:** 3
**Rating:** 4
**Confidence:** 4

**Summary:**

The authors constructed a simplified single-layer Transformer and successfully reproduced the emergence of linear truth encodings on a synthetic dataset. They proposed the Truth Co-occurrence Hypothesis (TCH), which posits that true statements tend to co-occur with other true statements, while false statements tend to co-occur with other false statements. The authors validated this hypothesis and revealed a two-stage dynamic in the model's training process: an initial rapid memorization phase, followed by the emergence of linear truth encoding.

**Questions:**

See Weaknesses

**Ethical Concerns:**

["NO or VERY MINOR ethics concerns only"]

**Final Justification:**

The rebuttal addressed my concerns. There are new experiments on LLMs and realistic data now.

**Limitations:**

Yes

**Quality:**

2

**Strengths And Weaknesses:**

Strengths

1、Truth Co-occurrence Hypothesis (TCH) is novel. The hypothesis offers a plausible explanation for the emergence of linear truth encodings, suggesting that the co-occurrence patterns of true and false statements drive the model to learn to distinguish between them.

2、Despite the model and the synthetic data are simple, the model effectively demonstrated key phenomena supporting the hypothesis, such as changes in the value matrix (Figure 1) and the model's ability to differentiate between true and false sequences (Figure 2).

3、The authors conducted experiments on the LLAMA3-8B model. Observing similar behaviors in this larger model.

Weaknesses

1. The use of a 1/2/3-layer Transformer model deviates significantly from the architecture of real-world LLMs, which typically have many layers. It is recommended to test the hypothesis in models with many layers to demonstrate its robustness and independence from model depth.

2. The synthetic dataset consists of simple quadruples, which fail to capture the complexity of real-world data distributions. In practice, datasets may involve more diverse subjects and partial truths.

3. In Section 5.2, the authors evaluated the LLAMA3-8B model using only 128 samples from the CounterFact dataset. This limited scope—both in terms of model and dataset—restricts the generalizability of their conclusions.

These weaknesses raise concerns about the practical validity of the paper, significantly decreasing my ratings.

---

> ### Author Rebuttal · Authors · 2025-07-30
>
> **We thank the reviewer for the thorough review and the encouragement to report empirical results on more realistic settings.**
> To address concerns about small models and synthetic data, we conducted two complementary experiments:
>
> 1. **Natural‑language facts.** We replicated our study by training fully‑parameterized Transformers on the CounterFact dataset.
> 2. **Large‑model trajectory.** We traced the entire training path of **Pythia‑6.9 B** to see whether the same dynamics emerge in the wild.
>
> ---
>
> ### Training dynamics on more realistic data
> Following your suggestion, we re‑ran the study in a more realistic setting—the **CounterFact** dataset [1], which contains simple factual statements in natural language. Please note that in contrast to our original synthetic setting, the examples from this dataset contain many different sentence structures and much variation that is characterized to natural language,
>
> We trained standard Transformer models with **2, 5, and 9 layers** (4 attention heads and one MLP block per layer) on a single relation, updating *all* parameters.
> Each training sample is a concatenation of two sequences, just as in the paper’s main experiments. To enforce the truth‑co‑occurrence hypothesis, the factuality of the two sequences is matched; in 5 % of the samples we corrupt the attribute by randomly swapping it with another attribute.
>
> The results reproduce the same **two‑phase learning pattern** seen in our synthetic setup:
>
> 1. **Phase 1 — rapid memorization**
>    Early in training the model places almost all probability mass on the memorized attribute for both true and false contexts, so the entropies are equally low (H\_true ≈ H\_false).
>
> 2. **Phase 2 — abrupt emergence of a linear encoding**
>    A few hundred steps later a strong linear signal appears in the penultimate token of the second sequence (AUC\_S2 ≈ 0.97).
>    At the same time H\_false shoots up while H\_true drops, exactly as predicted by theory.
>
> Because NeurIPS technical constraints, we summarize the average numbers across training phases.
>
> | Phase | Window&nbsp;(steps) | $\\mathrm{H_{true}}$ | $\\mathrm{H_{false}}$ | $\\mathrm{AUC_{S_1}}$ | $\\mathrm{AUC_{S_2}}$ |
> |:---|:---|:---:|:---:|:---:|:---:|
> | **2‑layer** — memorization | 0 – 1.2 k | 0.39 | 0.39 | 0.85 | 0.51 |
> | **2‑layer** — emergence | 1.4 k – 2 k → | ↓ 0.14 | ↑ 2.90 | 0.82 | **0.97** |
> | **5‑layer** — memorization | 0 – 750 | 0.41 | 0.40 | 0.70 | 0.52 |
> | **5‑layer** — emergence | 0.8 k – 1 k → | ↓ 0.12 | ↑ 2.31 | 0.98 | **0.97** |
> | **9‑layer** — memorization | 0 – 500 | 0.41 | 0.41 | 0.62 | 0.51 |
> | **9‑layer** — emergence | 0.6 k – 1 k → | ↓ 0.11 | ↑ 2.64 | 0.97 | **0.96** |
>
> Here, AUC\_S1 / AUC\_S2 refer to the Area Under the ROC Curve of a linear probe trained on
> &nbsp;&nbsp;&nbsp;&nbsp;• **S1**: the last token of the first sentence, or
> &nbsp;&nbsp;&nbsp;&nbsp;• **S2**: the penultimate token of the second sentence.
>
> In the 2‑layer model the memorization plateau lasts until ≈ 1.2 k steps; then the linear encoding emerges and H\_false rises, reducing loss on corrupted samples.
> The deeper models transition earlier—after only 500–750 steps—but follow the same trajectory.
> All metrics here are taken from the final layer. In the deeper networks the penultimate layer reaches comparable accuracy, while earlier layers stay near chance. Overall, these results confirm the two‑phase dynamics we predict in a more realistic setting.
>
> ---
>
> ### Evaluating checkpoints of a “real” LM
> To test whether the two‑phase dynamics also appear in a large model trained on open‑web data, we analyzed the **Pythia‑6.9 B** training checkpoints released by EleutherAI. Using the CounterFact dataset we construct each input by concatenating $K = 4$ factual statements whose preceding context is either entirely true or entirely false, mirroring our previous setup.
> For every checkpoint we measure three signals on the final statement:
>
> * **Memorization**: the percentage of cases where greedy decoding suceeds in completing the correct token.
> * **Uncertainty**: the entropy of the model’s full‑vocabulary distribution for predicting the last token (we record the difference in entropy between true and false context).
> * **Linear separability**: the accuracy of a linear probe trained to classify the truth value of the surrounding context.
>
> | step | $\\Delta H$ | memorization | probe&nbsp;AUC |
> |------:|-------:|---------------:|--------------:|
> | 0 | 0.001 | 0.000 | 0.383 |
> | 512 | 0.006 | 0.000 | 0.435 |
> | 1 000 | 0.005 | 0.006 | 0.467 |
> | 3 000 | 0.219 | 0.242 | 0.587 |
> | 5 000 | 0.217 | 0.435 | 0.648 |
> | 10 000 | 0.286 | 0.547 | 0.667 |
> | 20 000 | 0.355 | 0.655 | 0.754 |
> | 40 000 | 0.329 | 0.727 | 0.759 |
> | 60 000 | 0.421 | 0.772 | 0.802 |
> | 80 000 | 0.419 | 0.822 | 0.799 |
> | 100 000 | 0.479 | 0.835 | 0.818 |
> | 110 000 | 0.485 | 0.849 | 0.835 |
> | 120 000 | 0.536 | 0.842 | 0.835 |
> | 130 000 | 0.565 | 0.858 | 0.783 |
> | 143 000 | 0.518 | 0.875 | 0.831 |
>
> $\\Delta H$ denotes the entropy gap between matched prompt pairs presented with false versus true context. The memorization rate is the share of instances in which the model’s output distribution places the correct continuation token at the top‑1 position.
>
> **Findings.**
> * **Early training (≤ 1 k steps).** $\\Delta H \\approx 0$; the model memorizes indiscriminately.
> * **Mid training (3 k–80 k).** Memorization jumps, then plateaus, while $\\Delta H$ and probe accuracy climb steadily.
> * **Late training (≥ 80 k).** Entropy separation continues to widen even after memorization saturates, mirroring Phase 2 but over a longer horizon.
>
> This echoes the two‑phase pattern we observed in simpler experiments: an initial jump in memorization followed by a slower, steadier increase in entropy separation. There are, however, several differences. The second phase is more gradual, and classification and entropy increase even *before* memorization stabilizes. These differences likely stem from the model’s continual exposure to new facts during training, unlike our idealized setup where all facts are seen in a single gradient step. The modest final memorization and classification scores align with prior reports that the Pythia series is under‑trained relative to its capacity.
>
> **In sum**, both experiments corroborate the predicted two‑phase dynamics we observe.
>
> ---
>
> ### Additional models in the intervention experiments
> Following your concerns on the intervention experiments, we repeated the experiments on three more models with 1,024 examples (8× larger). Reported values, as in the main text, are **NLL deltas** between false and true contexts:
>
> * **Δ (Base F → Base T)** — change when switching the *base* model from false to true context.
> * **Δ (Base F → Interv F)** — change between *base* and *intervened* model on **false** context.
>
> | Model | false‑context length | Δ (Base F → Base T) | Δ (Base F → Interv F) |
> |:-----------------|:----------------:|:---------------:|:------------------:|
> | **Gemma2 7B** | FF | −4.25 | −3.36 |
> | | FFF | −3.68 | −3.09 |
> | | FFFF | −3.37 | −2.48 |
> | **Llama** | FF | −1.51 | −0.24 |
> | | FFF | −1.98 | −0.36 |
> | | FFFF | −2.18 | −0.31 |
> | **Qwen 2-7B** | FF | −0.30 | −0.15 |
> | | FFF | −0.38 | −0.22 |
> | | FFFF | −0.41 | −0.18 |
>
> Patterns are broadly consistent, though **Gemma** is more sensitive to false context than the others.
>
> ---
>
> ### Reference
> [1] Kevin Meng, David Bau, Alex Andonian, and Yonatan Belinkov. *Locating and Editing Factual Associations in GPT*, 2022.

---

> > ### Comment · Reviewer_za5X · 2025-08-04
> >
> > Thank you for your rebuttal. It addressed my concerns. I've revised my rating to positive.

---

### Official Review · Reviewer_aif5 · 2025-07-02

**Clarity:** 3
**Significance:** 3
**Originality:** 3
**Rating:** 5
**Confidence:** 3

**Summary:**

The paper looks into why and how LLMs learn a linear subspace encoding the truthfulness of statements, a phenomenon observed in many probing studies. They first proposed the TCH (truth co-occurance hypothesis) which means truth statements stay with truth statements in the data, while falsehoods stay with falsehoods. They further trained a toy transformer model to show the two stages in the learning process: the model first rapidly memorized the subject-attribute pairs, and then a linear separation gradually emerges between true and false statements. Finally, experiments with Llama-3-8B and CounterFact data show that after the false sentences exist, it reduces the likelihood of predicting true attributes. Also, when under intervention along the learned truth subspace, the model can be steered back toward correctness.

**Questions:**

- Why does the linear truth encoding peak in middle layers, e.g. layer 11 in Llama-3-8B? Is this consistent across model scales?
- Did you try any experiments on real models other than Llama-3-8B?

**Ethical Concerns:**

["NO or VERY MINOR ethics concerns only"]

**Final Justification:**

The authors tried their best to address most of my concerns in the rebuttal.

Me: The toy example can be too different from real LLMs.
Authors: Train real small-scale transformers on the CounterFact dataset again to show the results.

Me: There are Pythia models with model checkpoints that you can try.
Authors: Try new experiments with Pythia, show some similar observations, but also show a small difference (entropy increase before memorization stabilizes).

Me: Show me experiments with more model families.
Authors: Try Gemma and Qwen and show the results.

These additional experiments/contents addressed most of my original concerns. I will be happy to see this paper get accepted if these new experiments/results/discussion can be incorporated into the main sessions of this paper. I raised my score to 5.

**Limitations:**

yes

**Quality:**

2

**Strengths And Weaknesses:**

Strengths:

- The Truth Co-occurrence Hypothesis is novel and intuitive, and seems to be supported to some extent in the experiments.
- It offers a simple and interpretable mechanistic explanation for how the truth subspaces emerges in the training dynamics.

Weaknesses:

- The two-phrase finding from the toy transformer experiments seems not well verified on the real LLMs like Llama-3-8B. Only the linear intervention and the true/falsehoods co-occurance assumption was verified, not the two-phrase assumption.
- I think it's possible to use some of the open model checkpoints along the training trajectory to further verify the training dynamics of the real LLMs, for example, the Pythia models have the middle checkpoints available, which can be leveraged.
- The toy model and synthetic data are highly idealized (e.g., no MLPs, one attention head, orthogonal embeddings) and it's directly jumped to generalized to real LLMs, which is a bit suspicious. If small scale LLMs, like GPT-2 scale, can also be verified first, it would be more convincing.
- As different LLMs can have a bit different internal mechanisms, it will be better to test more than a single model family. Besides Llama3-8B, it's better to also show results from other models.

---

> ### Author Rebuttal · Authors · 2025-07-30
>
> **We thank the reviewer for their insightful suggestions and for encouraging additional large‑model experiments.**
> Below we address each point and indicate the text changes planned for the camera‑ready version.
>
> ---
>
> ### Theoretical modeling and idealized synthetic setting
> Our intention is to provide a rigorous characterization in a minimal setting: **if the phenomenon shows up even under these minimalist conditions, it becomes a plausible building block for the far richer dynamics of large‑scale models.** We emphasize that even our simplified objective (Eq. (2) after line 141) already contains a weight matrix $W$ trained *inside* a non‑linearity. This results in a non‑convex function that resembles a 2‑layer network, and its optimization process is known to be challenging to analyze, as accomplished in **Theorem 3**. Our analysis goes a step further and studies the *properties* of the learned weights (**Theorems 1 and 2**), which show the occurrence of the truth‑linear subspace.
>
> Please note that in the appendix we show that *releasing* the $K,Q,V$ and embeddings parameters (still one layer, but with nothing frozen) **retains the effect**, both the two-phases and the structure of the attention matrix (though when the embeddings are also trained, some of the structure disappears because they can also memorize). We will move this result into the main text for visibility.
>
> ---
>
> ### Training dynamics on more realistic data
> Following your suggestion, we re‑ran the study in a more realistic setting—the **CounterFact** dataset [1], which contains simple factual statements in natural language. Please note that in contrast to our original synthetic setting, the examples from this dataset contain many different sentence structures and much variation that is characterized to natural language,
>
> We trained standard Transformer models with **2, 5, and 9 layers** (4 attention heads and one MLP block per layer) on a single relation, updating *all* parameters.
> Each training sample is a concatenation of two sequences, just as in the paper’s main experiments. To enforce the truth‑co‑occurrence hypothesis, the factuality of the two sequences is matched; in 5 % of the samples we corrupt the attribute by randomly swapping it with another attribute.
>
> The results reproduce the same **two‑phase learning pattern** seen in our synthetic setup:
>
> 1. **Phase 1 — rapid memorization**
>    Early in training the model places almost all probability mass on the memorized attribute for both true and false contexts, so the entropies are equally low (H\_true ≈ H\_false).
>
> 2. **Phase 2 — abrupt emergence of a linear encoding**
>    A few hundred steps later a strong linear signal appears in the penultimate token of the second sequence (AUC\_S2 ≈ 0.97).
>    At the same time H\_false shoots up while H\_true drops, exactly as predicted by theory.
>
> Because NeurIPS technical constraints, we summarize the average numbers across training phases.
>
> | Phase | Window&nbsp;(steps) | $\\mathrm{H_{true}}$ | $\\mathrm{H_{false}}$ | $\\mathrm{AUC_{S_1}}$ | $\\mathrm{AUC_{S_2}}$ |
> |:---|:---|:---:|:---:|:---:|:---:|
> | **2‑layer** — memorization | 0 – 1.2 k | 0.39 | 0.39 | 0.85 | 0.51 |
> | **2‑layer** — emergence | 1.4 k – 2 k → | ↓ 0.14 | ↑ 2.90 | 0.82 | **0.97** |
> | **5‑layer** — memorization | 0 – 750 | 0.41 | 0.40 | 0.70 | 0.52 |
> | **5‑layer** — emergence | 0.8 k – 1 k → | ↓ 0.12 | ↑ 2.31 | 0.98 | **0.97** |
> | **9‑layer** — memorization | 0 – 500 | 0.41 | 0.41 | 0.62 | 0.51 |
> | **9‑layer** — emergence | 0.6 k – 1 k → | ↓ 0.11 | ↑ 2.64 | 0.97 | **0.96** |
>
> Here, AUC\_S1 / AUC\_S2 refer to the Area Under the ROC Curve of a linear probe trained on
> &nbsp;&nbsp;&nbsp;&nbsp;• **S1**: the last token of the first sentence, or
> &nbsp;&nbsp;&nbsp;&nbsp;• **S2**: the penultimate token of the second sentence.
>
> In the 2‑layer model the memorization plateau lasts until ≈ 1.2 k steps; then the linear encoding emerges and H\_false rises, reducing loss on corrupted samples.
> The deeper models transition earlier—after only 500–750 steps—but follow the same trajectory.
> All metrics here are taken from the final layer. In the deeper networks the penultimate layer reaches comparable accuracy, while earlier layers stay near chance. Overall, these results confirm the two‑phase dynamics we predict in a more realistic setting.
>
> ---
>
> ### Evaluating checkpoints of a “real” LM
> To test whether the two‑phase dynamics also appear in a large model trained on open‑web data, we analyzed the **Pythia‑6.9 B** training checkpoints released by EleutherAI. Using the CounterFact dataset we construct each input by concatenating $K = 4$ factual statements whose preceding context is either entirely true or entirely false, mirroring our previous setup.
> For every checkpoint we measure three signals on the final statement:
>
> * **Memorization**: the percentage of cases where greedy decoding suceeds in completing the correct token.
> * **Uncertainty**: the entropy of the model’s full‑vocabulary distribution for predicting the last token (we record the difference in entropy between true and false context).
> * **Linear separability**: the accuracy of a linear probe trained to classify the truth value of the surrounding context.
>
> | step | $\\Delta H$ | memorization | probe&nbsp;AUC |
> |------:|-------:|---------------:|--------------:|
> | 0 | 0.001 | 0.000 | 0.383 |
> | 512 | 0.006 | 0.000 | 0.435 |
> | 1 000 | 0.005 | 0.006 | 0.467 |
> | 3 000 | 0.219 | 0.242 | 0.587 |
> | 5 000 | 0.217 | 0.435 | 0.648 |
> | 10 000 | 0.286 | 0.547 | 0.667 |
> | 20 000 | 0.355 | 0.655 | 0.754 |
> | 40 000 | 0.329 | 0.727 | 0.759 |
> | 60 000 | 0.421 | 0.772 | 0.802 |
> | 80 000 | 0.419 | 0.822 | 0.799 |
> | 100 000 | 0.479 | 0.835 | 0.818 |
> | 110 000 | 0.485 | 0.849 | 0.835 |
> | 120 000 | 0.536 | 0.842 | 0.835 |
> | 130 000 | 0.565 | 0.858 | 0.783 |
> | 143 000 | 0.518 | 0.875 | 0.831 |
>
> $\\Delta H$ denotes the entropy gap between matched prompt pairs presented with false versus true context. The memorization rate is the share of instances in which the model’s output distribution places the correct continuation token at the top‑1 position.
>
> **Findings.**
> * **Early training (≤ 1 k steps).** $\\Delta H \\approx 0$; the model memorizes indiscriminately.
> * **Mid training (3 k–80 k).** Memorization jumps, then plateaus, while $\\Delta H$ and probe accuracy climb steadily.
> * **Late training (≥ 80 k).** Entropy separation continues to widen even after memorization saturates, mirroring Phase 2 but over a longer horizon.
>
> This echoes the two‑phase pattern we observed in simpler experiments: an initial jump in memorization followed by a slower, steadier increase in entropy separation. There are, however, several differences. The second phase is more gradual, and classification and entropy increase even *before* memorization stabilizes. These differences likely stem from the model’s continual exposure to new facts during training, unlike our idealized setup where all facts are seen in a single gradient step. The modest final memorization and classification scores align with prior reports that the Pythia series is under‑trained relative to its capacity.
>
> **In sum**, both experiments corroborate the predicted two‑phase dynamics we observe.
>
> ---
>
> ### Additional models in the intervention experiments
> Following your concerns on the intervention experiments, we repeated the experiments on three more models with 1,024 examples (8× larger). Reported values, as in the main text, are **NLL deltas** between false and true contexts:
>
> * **Δ (Base F → Base T)** — change when switching the *base* model from false to true context.
> * **Δ (Base F → Interv F)** — change between *base* and *intervened* model on **false** context.
>
> | Model | false‑context length | Δ (Base F → Base T) | Δ (Base F → Interv F) |
> |:-----------------|:----------------:|:---------------:|:------------------:|
> | **Gemma2 7B** | FF | −4.25 | −3.36 |
> | | FFF | −3.68 | −3.09 |
> | | FFFF | −3.37 | −2.48 |
> | **Llama** | FF | −1.51 | −0.24 |
> | | FFF | −1.98 | −0.36 |
> | | FFFF | −2.18 | −0.31 |
> | **Qwen 2-7B** | FF | −0.30 | −0.15 |
> | | FFF | −0.38 | −0.22 |
> | | FFFF | −0.41 | −0.18 |
>
> Patterns are broadly consistent, though **Gemma** is more sensitive to false context than the others.
>
> ---
>
> ### References
> Meng, K., Bau, D., Andonian, A., & Belinkov, Y. *Locating and Editing Factual Associations in GPT*, 2022.
>
> ---
>
> Regarding your question on layer 11, this is in line with previous work that indicates that many human-interpretable concepts are "maximized" in middle layers. We agree that understanding why it is maximized in that particular layer is very interesting, but think that this is a different question than the focus of this paper, and therefore leave it for a future work.
>
> Once again, we appreciate the reviewer’s constructive feedback and believe these additional results strengthen the paper’s empirical grounding.

---

> ### Comment · Reviewer_aif5 · 2025-08-04
>
> Thanks for the response with rich content! I am glad that the authors tried their best to address most of my concerns.
>
> Me: The toy example can be too different from real LLMs.
> Authors: Train real small-scale transformers on the CounterFact dataset again to show the results.
>
> Me: There are Pythia models with model checkpoints that you can try.
> Authors: Try new experiments with Pythia, show some similar observations, but also show a small difference (entropy increase before memorization stabilizes).
>
> Me: Show me experiments with more model families.
> Authors: Try Gemma and Qwen and show the results.
>
> These additional experiments/contents addressed most of my original concerns. I will be happy to see this paper get accepted if these new experiments/results/discussion can be incorporated into the main sessions of this paper, which will fundamentally makes this paper much more solid. However, there are still a lot of experiment details, not shown in the tables above, that should be added later on. I tends to trust the authors will do their best in the camera ready version to incorporate these new contents with all the details. If so, I will be happy to raise my scores to accept this paper.

---

> > ### Author Response · Authors · 2025-08-04
> > **Response**
> >
> > Thank you for the response! We greatly appreciate your encouraging follow-up.
> >
> > We will incorporate these new results, together with a full description of the experimental setup, into the camera ready version. We agree that they significantly help in bridging the gap between our simplified theoretical model and real-world language models. Please let us know if any further clarification would help.

---

### Official Review · Reviewer_CoF5 · 2025-07-02

**Clarity:** 3
**Significance:** 3
**Originality:** 4
**Rating:** 5
**Confidence:** 3

**Summary:**

The paper explores the emergence of linear separable truth encoding in Transformer models. They explore a toy setting with two token pairs where if the first token pair follows a 'ground truth' distribution, the second token pair is likely to also follow that distribution (replicating the Truth Co-occurrence Hypothesis: in naturally occurring text,
true statements are statistically more likely to co-occur with other true statements, and falsehoods with other falsehoods.). They show that the transformer first learns the ground truth mapping and only later learns to apply it to the second token pair only in the case of a first ground truth pair.

**Questions:**

**Questions:**

- In Figure 2: would it make sense to plot averaged values over several inputs?
- Are sequences always 4 tokens long? What happens if you make them longer?
- Could you add a few examples of your dataset? How do you define a^* is it just 2*s? Does it matter how you define it or can it be an arbitrary mapping?
- In (1) shouldn't the prefix be x=(s1 a1 s2)?
- Why do you switch between s,a and x,y? why not just stick with s and a?


**Suggestions:**

- explain shortly what e_x, u_{g(x)} etc is in figure caption so the figure+caption is self contained. Maybe also mark which block is which, since it's hard to infer just by looking at the figure.
- add axes labels in Figure 2
- in the setup in line 137 you could add the symbols e, p, u when after embedding, positional embedding, and unembedding respectively to have them clearly defined
- add definition for F in eq (2)
- add definition of g when symbol is first used
- be more consistent with what you call things: for example you define a sequence with subjects and attributes, but later in line 155 you call then input and label
- maybe for Figure 4 c) add that these are learned embeddings since in the toy example they are pre defined

**Ethical Concerns:**

["NO or VERY MINOR ethics concerns only"]

**Limitations:**

The authors adequately discuss limitations and list interesting future directions

**Paper Formatting Concerns:**

No concerns

**Quality:**

3

**Strengths And Weaknesses:**

**Strengths:**

I think studying the Co-occurrence Hypothesis for truthful statements in a toy example where one can directly control conditions is a good approach.
The authors motivate their approach well and set a clear goal/focus. The paper is set up in a well structured way.
The contribution is to the best of my knowledge novel and significant.
They adequalely cover and acknowledge previous work.


**Weaknesses:**

The authors could be a bit more diligent with defining symbols to make it easier for readers to follow. Some Figure captions could be improved for clarity.

minor:

- You should not use LM in the abstract without defining it (which you could easily do since you already mention large language models)
- line 68 typo by -> be
- line 161: verb missing in '(though we still some lower confidence spikes...'

---

> ### Author Rebuttal · Authors · 2025-07-30
>
> **We thank the reviewer for their positive assessment and for highlighting both the paper’s structure and novelty.**
> Below we answer the specific questions and note the text changes we will make in the camera‑ready version.
>
> ---
>
> ### General edits
> We appreciate the comments on typos and the need to improve the captions and results descriptions. We will incorporate these in the final version.
>
> ---
>
> ### Figure&nbsp;2
> Yes—the patterns are reproducible. The particular values that are *arg‑maxed* naturally differ between examples where the gold token changes, *but* the distribution becomes flattened when conditioned on false contexts, as shown by the aggregated entropy numbers we report. We will convey this more clearly in the final version.
>
> ---
>
> ### Example length
> In the experiments included in this paper, the sequence length is indeed fixed; this makes the theoretical analysis easier. However, in preliminary runs we inserted extra padding tokens at random, and this did **not** change the empirical picture.
>
> ---
>
> ### Mapping (subject ↔ attribute)
> The relation between subjects and their attributes is defined by a random permutation that depends only on the random seed. The results are not sensitive to that seed.
> Please also refer to the additional experiment described in the response to the other reviewers (technical constraints at NeurIPS prevented us from attaching a general response). There we reproduce similar dynamics in a model trained on *natural‑language* data satisfying the truth‑co‑occurrence hypothesis. In that setting the examples no longer have a rigid structure, yet we observe the same phenomena as in the synthetic data.
>
> ---
>
> ### Eq.&nbsp;1
> Yes—thank you for pointing this out.
>
> ---
>
> ### Notation
> Thank you as well; we will standardize the notation for the subject and attribute throughout.
>
> ---
>
> Once again, we are grateful for the reviewer’s careful reading and helpful suggestions, and we look forward to refining the paper accordingly.

---

> > ### Comment · Reviewer_CoF5 · 2025-08-01
> >
> > Thanks for the reply.
> >
> > I feel a bit less confident in my assessment after reading the other reviewers criticism.
> >
> > I still like the paper though, I think the authors adequately discuss the limitations of their toy example and do not over claim generalizations. I think it is fine to have a paper focused on a toy example and have future work building on that and will therefore keep my score.

---

> ### Author Response · Authors · 2025-08-02
> **Response**
>
> Thank you for the follow-up and for maintaining your positive assessment of our work! Please note we aimed to answer the other reviewers’ concerns in detail in the corresponding responses, providing new results that substantiate our findings in more realistic settings.

---

### Official Review · Reviewer_ZrWv · 2025-07-03

**Clarity:** 3
**Significance:** 2
**Originality:** 3
**Rating:** 3
**Confidence:** 4

**Summary:**

The authors propose a simple theory that aims to explain, in part, how language models learn “truth” directions. They introduce the Truth-Co-occurrence Hypothesis (TCH)—namely, that true statements tend to appear near other true statements, and likewise for falsehoods. To test this idea, they build a TCH-driven synthetic dataset in which truthful or untruthful statements always occur together, then train a single-layer transformer (with several components frozen) on that data. Internally, the model learns to classify truthful versus untruthful statements linearly by their norms. The authors also explore whether a similar mechanism emerges in larger language models under a different setup. The paper offers an interesting perspective and illustrative toy experiments showing how a model might acquire a linear truth direction; however, I still feel the findings remain largely detached from real-world LMs (see my comments below).

**Questions:**

See my comments above.

**Ethical Concerns:**

["NO or VERY MINOR ethics concerns only"]

**Final Justification:**

I am raising my scores since the newer experiment results make me more comfortable of the toy experiment result included in the draft. I am increasing my score only slightly since, without revision, I still feel like the paper makes a broader claim which is not that well-supported by experiments included.

**Limitations:**

See my comments above.

**Paper Formatting Concerns:**

No.

**Quality:**

2

**Strengths And Weaknesses:**

**Strengths**

* TCH is a simple yet interesting hypothesis.
* The toy setting is engaging, and the observation that the learned norms encode a linear direction that separates truthful from untruthful statements is likewise noteworthy.
* The paper is well written and easy to follow.

**Weaknesses**

* **The *main* toy setting imposes interesting yet unrealistic constraints that may render the norm-separation discovery trivial.** The toy model essentially freezes everything except *a single weight matrix*. Given that truthful data are more frequent and the loss is unbounded (i.e., pushed toward more negative values), it is unsurprising that some truthful subspace in the value matrix (i.e., certain column spans) receives more frequent gradient updates, driving its norms out of distribution. Although one could argue the learned value matrix is full-rank, the small vocabulary means the meaningful updates are effectively low-rank. In line 272, the authors even note that this finding “disappears when learning the KV matrices instead of using fixed attention.” I suspect this is because the pressure on norms vanishes once more parameters are trainable. Overall, uniform attention strikes me as an unrealistic assumption.

  *P.S.* Proving a hypothesis with toy settings is fine if all claims are stated with explicit assumptions, but I remain unconvinced that this is true here.

* **The follow-up study on the 8B LM does not test the training dynamics observed in the toy example at scale.** The toy experiment purports to show that TCH explains the emergence of a linear truth direction; the 8B study instead demonstrates that an LM is biased by untruthful context and that a steering vector can restore truthfulness—both fairly well-known results. I would urge the authors to scale up their training-dynamics analysis (e.g., loosen the constraints or train larger transformers on more realistic, partially corrupted factual corpora).

---

> ### Author Rebuttal · Authors · 2025-07-30
>
> **We sincerely thank the reviewer for their thoughtful and constructive feedback, and we appreciate the opportunity to clarify our work.**
> We address each point in turn below.
>
> ---
>
> ### Regarding the sentence at line 272
> This sentence was unfortunately a remnant from a preliminary version and doesn't accurately describe our findings. We apologize for the confusion. **Frozen attention is *not* necessary for reconstructing the pattern in Fig. 1.** Indeed, in **Appendix C** (“Bridging the gap between the fully‑trainable model and the toy model”, line 852) we report the trainable‑attention results: the heat‑map in **Fig. 10a** retains the same pattern we see in Fig 1 in the main text, while **Fig. 10b** shows that if embeddings are also learned, part of the memorization moves there and the pattern weakens. *The two phases dynamics and emergence also remain when updating all parameters*; this is how figure 3 was generated. We will clarify this in the main text and fix line 272. Please also see “Training dynamics” below.
>
> ---
>
> ### Uniform‑attention head and other assumptions
> We recognize that idealizations such as uniform, fixed attention (together with a single‑layer architecture) are not faithful to production LMs. They are, however, *standard tools* in mechanistic‑transformer theory because they strip away distractions and let us isolate one concrete route by which a linear “truth” direction can form. Similar works that study different properties of self‑attention also make simplifying assumptions, such as fixing the attention weights and training only a single weight matrix—see, for example, [1] on implicit bias and [2] on benign overfitting in self‑attention.
>
> Our intention is therefore to provide a rigorous characterization in a minimal setting: **if the phenomenon shows up even under these minimalist conditions, it becomes a plausible building block for the far richer dynamics of large‑scale models.** We emphasize that even our simplified objective (Eq. (2) after line 141) already contains a weight matrix $W$ trained *inside* a non‑linearity. This results in a non‑convex function that resembles a 2‑layer network, and its optimization process is known to be challenging to analyze, as accomplished in **Theorem 3**. Our analysis goes a step further and studies the *properties* of the learned weights (**Theorems 1 and 2**), which show the occurrence of the truth‑linear subspace.
>
> We also remark that attention is typically uniform at initialization in high dimension, and that the value matrix usually receives more gradient signal than the key‑query matrix initially [4, 5], making our assumption relevant for the *initial phase* of training. After the value matrix has learned the structure in Eqs. (4–5)—which already gives good performance—we found empirically that when $\\rho = 1$ (only true sequences), the key‑query matrix tends to learn a positional pattern that avoids attending to the *x* and *y* tokens from *x′*. We now have an analysis of how this happens via a gradient step on this matrix: the model with uniform attention over‑ (resp. under-)estimates the probability of $y′ = g(x)$ (resp. $y′ = y$) on true sequences, hence the gradient places smaller attention on the *x*, *y* tokens, using positional embeddings. We will happily include this proof in the appendix.
>
> ---
>
> ### Small vocabulary and “effectively low rank”
> Please note that in the appendix (**Figure 7**) we re‑ran the toy experiment with vocabulary sizes 128, 512, and 4096 and different hidden‑state dimensions. In *all* cases, Linear‑probe AUC stays above 0.86. Furthermore, the positive‑classification margin guaranteed in **Theorem 2** does *not* depend on vocabulary size; we will point this out directly after the theorem.
>
> ---
>
> ### Training dynamics on more realistic data
> Following your suggestion, we re‑ran the study in a more realistic setting—the **CounterFact** dataset [3], which contains simple factual statements in natural language, with variations in sentence structure.
>
> We trained standard Transformer models with **2, 5, and 9 layers** (4 attention heads and one MLP per layer) on a single relation, updating *all* parameters.
> Each training sample is a concatenation of two sequences, just as in the paper’s main experiments. To enforce the truth‑co‑occurrence hypothesis, the factuality of the two sequences is matched; in 5 % of the samples we corrupt the attribute by randomly swapping it with another attribute.
>
> The results reproduce the same **two‑phase learning pattern** seen in our synthetic setup:
>
> 1. **Phase 1 — rapid memorization**
>    Early in training the model places almost all probability mass on the memorized attribute for both true and false contexts, so the entropies are equally low (H\_true ≈ H\_false).
>
> 2. **Phase 2 — abrupt emergence of a linear encoding**
>    A few hundred steps later a strong linear signal appears in the penultimate token of the second sequence (AUC\_S2 ≈ 0.97).
>    At the same time H\_false shoots up while H\_true drops, exactly as predicted by theory.
>
> Because NeurIPS technical constraints, we summarize the average numbers across training phases:
>
> | Phase | Window&nbsp;(steps) | $\\mathrm{H_{true}}$ | $\\mathrm{H_{false}}$ | $\\mathrm{AUC_{S_1}}$ | $\\mathrm{AUC_{S_2}}$ |
> |:---|:---|:---:|:---:|:---:|:---:|
> | **2‑layer** — memorization | 0 – 1.2 k | 0.39 | 0.39 | 0.85 | 0.51 |
> | **2‑layer** — emergence | 1.4 k – 2 k → | ↓ 0.14 | ↑ 2.90 | 0.82 | **0.97** |
> | **5‑layer** — memorization | 0 – 750 | 0.41 | 0.40 | 0.70 | 0.52 |
> | **5‑layer** — emergence | 0.8 k – 1 k → | ↓ 0.12 | ↑ 2.31 | 0.98 | **0.97** |
> | **9‑layer** — memorization | 0 – 500 | 0.41 | 0.41 | 0.62 | 0.51 |
> | **9‑layer** — emergence | 0.6 k – 1 k → | ↓ 0.11 | ↑ 2.64 | 0.97 | **0.96** |
>
> Here, AUC\_S1 / AUC\_S2 refer to the Area Under the ROC Curve of a linear probe trained on
> &nbsp;&nbsp;&nbsp;&nbsp;• **S1**: the last token of the first sentence, or
> &nbsp;&nbsp;&nbsp;&nbsp;• **S2**: the penultimate token of the second sentence.
>
> In the 2‑layer model the memorization plateau lasts until ≈ 1.2 k steps; then the linear encoding emerges and H\_false rises, reducing loss on corrupted samples.
> The deeper models transition earlier—after only 500–750 steps—but follow the same trajectory.
> All metrics are taken from the final layer. In the deeper networks the penultimate layer reaches comparable accuracy, while earlier layers stay near chance. Overall, these results confirm the two‑phase dynamics we predict in a more realistic setting.
>
> ---
>
> ### Evaluating checkpoints of a “real” LM
> To test whether the two‑phase dynamics also appear in a large model trained on open‑web data, we analyzed the **Pythia‑6.9 B** training checkpoints released by EleutherAI. Using the CounterFact dataset we construct each input by concatenating $K = 4$ factual statements whose preceding context is either entirely true or entirely false, mirroring our previous setup.
> For every checkpoint we measure three signals on the final statement:
>
> * **Memorization**: the percentage of cases where greedy decoding suceeds in completing the correct token.
> * **Uncertainty**: the entropy of the model’s full‑vocabulary distribution for predicting the last token (we record the difference in entropy between true and false context).
> * **Linear separability**: the accuracy of a linear probe trained to classify the truth value of the surrounding context.
>
> | step | $\\Delta H$ | memorization | probe&nbsp;AUC |
> |------:|-------:|---------------:|--------------:|
> | 0 | 0.001 | 0.000 | 0.383 |
> | 512 | 0.006 | 0.000 | 0.435 |
> | 1 000 | 0.005 | 0.006 | 0.467 |
> | 3 000 | 0.219 | 0.242 | 0.587 |
> | 5 000 | 0.217 | 0.435 | 0.648 |
> | 10 000 | 0.286 | 0.547 | 0.667 |
> | 20 000 | 0.355 | 0.655 | 0.754 |
> | 40 000 | 0.329 | 0.727 | 0.759 |
> | 60 000 | 0.421 | 0.772 | 0.802 |
> | 80 000 | 0.419 | 0.822 | 0.799 |
> | 100 000 | 0.479 | 0.835 | 0.818 |
> | 110 000 | 0.485 | 0.849 | 0.835 |
> | 120 000 | 0.536 | 0.842 | 0.835 |
> | 130 000 | 0.565 | 0.858 | 0.783 |
> | 143 000 | 0.518 | 0.875 | 0.831 |
>
> $\\Delta H$ denotes the entropy gap between matched prompt pairs presented with false versus true context. The memorization rate is the share of instances in which the model’s output distribution places the correct continuation token at the top‑1 position.
>
> **Findings.**
> * **Early training (≤ 1 k steps).** $\\Delta H \\approx 0$; the model memorizes indiscriminately.
> * **Mid training (3 k–80 k).** Memorization jumps, then plateaus, while $\\Delta H$ and probe accuracy climb steadily.
> * **Late training (≥ 80 k).** Entropy separation continues to widen even after memorization saturates, mirroring Phase 2 but over a longer horizon.
>
> This echoes the two‑phase pattern we observed in simpler experiments: an initial jump in memorization followed by a slower, steadier increase in entropy separation. There are, however, several differences. The second phase is more gradual, and classification and entropy increase even *before* memorization stabilizes. These differences likely stem from the model’s continual exposure to new facts during training, unlike our idealized setup where all facts are seen in a single gradient step. The modest final memorization and classification scores align with prior reports that the Pythia series is under‑trained relative to its capacity.
>
> **In sum**, both experiments corroborate the predicted two‑phase dynamics we observe.
>
> ### References
> [1] Vasudeva, Bhavya, Puneesh Deora, and Christos Thrampoulidis. *Implicit bias and fast convergence rates for self‑attention*, 2024.
> [2] Magen, Roey, *et al.* *Benign overfitting in single‑head attention*, 2025.
> [3] Kevin Meng, David Bau, Alex Andonian, and Yonatan Belinkov. *Locating and Editing Factual Associations in GPT*, 2022.
> [4] Li, Li, and Risteski. *How Do Transformers Learn Topic Structure: Towards a Mechanistic Understanding*, ICML 2023.
> [5] Bietti, et al. *Birth of a Transformer: A Memory Viewpoint*, NeurIPS 2023.

---

> > ### Comment · Reviewer_ZrWv · 2025-08-04
> > **Thank you for your response.**
> >
> > With additional experiments, I am slightly more confident of the toy experiment results. Thanks.

---

> > > ### Author Response · Authors · 2025-08-04
> > > **Response**
> > >
> > > Thank you for your follow-up and for reviewing the additional experiments. We’re glad they increased your confidence in the toy-experiment findings. We will fold these results---and their full experimental setup details---into the camera ready. If you have any other questions or would like further clarification, please let us know.

---

### Note · Authors · 2025-08-11

We thank the reviewers and AC for their constructive engagement in the discussion period, and for recognizing the strengths of our work -- namely, the novelty of the Truth Co-occurrence Hypothesis (TCH), the clarity and structure of the paper, and the value of a minimal mechanistic setting for isolating the emergence of a "truth direction" in language models.

In response to concerns about realism and generalizability, we reported three new lines of experiments: (1) Bridging the toy vs. realistic gap: fully-parameterized Transformers (2, 5, 9 layers) trained on the CounterFact dataset reproduced the predicted two-phase dynamics despite natural-language variability; (2) Large-model training dynamics: analyzing Pythia-6.9B checkpoints revealed similar qualitative two-phase pattern in a model trained on open-web data; (3) Cross-model robustness: intervention effects confirmed on Gemma 2-7B, Qwen 2-7B, and Llama models. The reviewers’ follow-up comments indicate increased confidence and support, reflected by both their response and the significant rise in their assigned scores.

With these additions, the paper now offers both a rigorous minimal mechanistic account of how linear truth encodings can emerge, and empirical evidence across synthetic, small-scale, and large-scale settings. We believe this bridges theory and practice in a way that is novel, interpretable, and broadly relevant to mechanistic interpretability and factuality research. We appreciate the reviewers' positive response to our rebuttal. We believe that their suggestions significantly strengthen our work, and look forward to incorporating them in the camera-ready.

---

### Decision · Program_Chairs · 2025-09-17

**Decision:**

Accept (poster)

**Comment:**

The paper studies the emergence of linear truth encodings (linear subspaces that separate true and false statements in the internal representations of a trained model). The paper studies a simple one-layer transformer model trained on a data distribution where true facts co-occur with other true facts more often than false facts. In this toy setting they show that the model learns to classify truthful versus untruthful statements linearly by their norms in a two-phase mechanism. The authors corroborate their findings on larger pre-trained models experimentally.

All reviewers agreed that the studied hypothesis is interesting and the toy setup is a clean abstraction of it. There were several concerns raised initially: (1) the simplicity of the Transformer model having only one linear trainable parameter, and the trends not being as clear when more parameters were trained, (2) lack of empirical evidence on pre-trained model matching the two-phase dynamics proposed in the theory, (3) limited evaluation on different sized models, and more realistic datasets. Apart from (1), the other concerns were addressed by the authors during the rebuttal with extensive experiments.

While the theoretical setup remain too idealized, the empirical validation of the phenomenon is good, and the core hypothesis is simple and plausible. Therefore I lean towards accept. I encourage the authors to include the new experiments done during the rebuttal period and improve the exposition to address reviewer concerns.